

# Greenland's firn responds more to warming than to cooling

Megan Thompson-Munson[1,2], Jennifer E. Kay[1,2], Bradley R. Markle[3,4]

[1]Cooperative Institute for Research in Environmental Sciences, University of Colorado, Boulder, CO 80309, USA
[2]Department of Atmospheric and Oceanic Sciences, University of Colorado, Boulder, CO 80309, USA
[3]Institute of Arctic and Alpine Research, University of Colorado, Boulder, CO 80309, USA
[4]Department of Geological Sciences, University of Colorado, Boulder, CO 80309, USA

*Correspondence to*: Megan Thompson-Munson (megan.thompson-munson@colorado.edu)

**Abstract.** The porous layer of snow and firn on the Greenland Ice Sheet stores meltwater and limits sea level rise. This buffer
is threatened in a warming climate. To better understand the nature and timescales of firn's response to air temperature change,
we use a physics-based model to assess the effects of atmospheric warming and cooling on firn air content. We identify an
asymmetric response of Greenland's firn to air temperature: firn loses more air content due to warming compared to the amount
gained from commensurate cooling. In dry firn, this asymmetry is driven by the highly nonlinear relationship between
temperature and firn compaction, as well as the dependency of thermal conductivity on the composition of the firn. The
influence of liquid water accentuates this asymmetry. In wet firn areas, melt increases nonlinearly with atmospheric warming,
thus enhancing firn refreezing and further warming the snowpack through increased latent heat release. Our results highlight
the vulnerability of firn to temperature change and demonstrate that firn air content is more efficiently depleted than generated.
This asymmetry in the temperature–firn relationship may contribute to the overall asymmetric mass change of the Greenland
Ice Sheet in a changing climate across many timescales.



## 1 Introduction

The Greenland Ice Sheet exhibits asymmetric mass changes closely tied to temperature changes (Alley et al., 2010). Specifically, the ice sheet gradually gains mass in periods of cooling and rapidly loses mass in periods of warming (Alley et al., 2010; Broecker and van Donk, 1970). Mass change occurs through the combination of ice discharge across grounding lines and mass fluxes at the ice sheet surface. The energy balance at the surface controls the production of meltwater that eventually runs off to contribution to sea level rise.

While atmospheric warming drives meltwater runoff on Greenland (Hanna et al., 2008), not all meltwater generated at the surface runs off into the ocean. A layer of porous snow and firn covers most of the ice sheet and partly buffers the contribution to sea level rise by storing meltwater (Harper et al., 2012). Firn is the transitional material between fresh snow and glacial ice, and its porous nature allows for the retention of meltwater in pore space (Pfeffer et al., 1991). When melting occurs in areas covered by firn, the meltwater can percolate into the snowpack and remain liquid within a firn aquifer (Forster

et al., 2014) or refreeze within the pore space (Pfeffer et al., 1991). In either case, firn can store meltwater if pore space is available, and it can limit the ice sheet's contribution to sea level rise (Harper et al., 2012).

      The efficiency of firn to retain meltwater depends on firn properties such as air content and density. Changes to firn air content and firn density are in part driven by compaction, in which overburden stress compresses firn (MacFerrin et al., 2022). Air temperature and accumulation have strong controls on the compaction rate (Herron and Langway, 1980) and lead

to temporal variations in both density (e.g., Vandecrux et al., 2018) and air content (e.g., Thompson-Munson et al., 2023). Compaction is the primary means of densification in the colder dry areas of Greenland, but the presence of meltwater in warmer areas introduces additional processes that have been shown to alter firn properties (Kuipers Munneke et al., 2015). When meltwater fills and refreezes within the firn it releases latent heat that increases the firn temperature and enhances the compaction rate (Pfeffer and Humphrey, 1996). Thus, firn air content is reduced both by the occupation of pore space by water

and the intensified compaction from the latent heat released during the refreezing of that meltwater.

      Recent observed warming in Greenland demonstrates the importance of firn–climate interactions. Greenland has experienced rising temperatures (Trusel et al., 2018), and the influence of this atmospheric warming on the firn layer has been observed. For example, meltwater percolation and refreezing have caused recent firn warming in several regions of the ice sheet (Polashenski et al., 2014; Charalampidis et al., 2016; Humphrey et al., 2012). In the high-melt areas of southeast

Greenland where firn has warmed enough to substantially reduce its cold content, reduced refreezing of meltwater has led to the expansion of firn aquifers (Horlings et al., 2022). Atmospheric warming and enhanced meltwater production have also increased the firn–ice content through the refreezing of meltwater within the firn (de la Peña et al., 2015; Graeter et al., 2018). With enough refreezing, discontinuous ice lenses and layers can grow into near-surface, low-permeability ice slabs that block access to deeper pore space (Machguth et al., 2016) and amplify runoff (MacFerrin et al., 2019). Recent high-melt years are

responsible for the expansion and thickening of these ice slabs (Jullien et al., 2023) as well as the resultant reduced permeability that inhibits meltwater retention in the firn (Culberg et al., 2021). These climate-driven changes have caused a reduction of



Greenland's firn air content since the early 2000s (Thompson-Munson et al., 2023), especially along the western margin of the ice sheet (Vandecrux et al., 2019).

Our current climate offers the opportunity to observe firn–climate interactions influenced by a warming atmosphere.
However, advancing our understanding of these interactions in a cooling climate using modern observations is not possible. Yet, firn behavior in a cooling climate is equally important to understand and is especially interesting to contrast with firn behavior in a warming climate. Thus, the primary goal of this work is to contrast the processes affecting the firn response to atmospheric warming and cooling using modern modeling tools. Our work provides additional insights into firn's influence on ice sheet mass changes in response to air temperature change that are not available from the modern instrumental record.
From our modeling, we find that firn processes contribute to the overall asymmetry of the Greenland Ice Sheet response to atmospheric warming and cooling. Specifically, temperature–firn interactions amplify pore space loss more in a warming climate than they amplify pore space gain in a cooling climate. This asymmetric air temperature influence on firn pore space has important implications for Greenland mass loss and gain.

## 2 Methods

### 2.1 Firn model description

We use a single-column, multi-layer, physics-based model called SNOWPACK (Bartelt and Lehning, 2002; Lehning et al., 2002a, b) to simulate time-evolving firn properties across the entire Greenland Ice Sheet. Previous studies have used SNOWPACK to model snow and firn processes in both Antarctica (Dunmire et al., 2020; Keenan et al., 2021; van Wessem et al., 2021; Wever et al., 2022; Maclennan et al., 2023; Banwell et al., 2023) and Greenland (Groot Zwaaftink et al., 2013; Van
Tricht et al., 2016; Steger et al., 2017; Izeboud et al., 2020; Dunmire et al., 2021; Thompson-Munson et al., 2023). SNOWPACK uses near-surface atmospheric conditions as input and operates within a Lagrangian framework in which a new layer is added when precipitation occurs. The density of the new layer is calculated from the atmospheric forcing and considers the effects of blowing snow on compaction (Keenan et al., 2021; Wever et al., 2022). Within the simulated firn column, densification is calculated with a constitutive relationship between stress and strain in snow (Bartelt and Lehning, 2002;
Lehning et al., 2002a).

SNOWPACK contains its own surface energy balance scheme, which is used to calculate the upper boundary condition for the firn temperature. Unlike semi-empirical firn models used in similar studies (e.g., Kuipers Munneke et al., 2015), SNOWPACK does not use a positive degree day parameterization for calculating melt. Instead, it uses an energy balance model to calculate melt in a way that incorporates processes occurring throughout the firn column in addition to those at the
surface (Wever et al., 2014, 2015, 2016). To represent the vertical flow of liquid water throughout the firn, we apply a bucket scheme. While other more complex approaches to governing meltwater percolation exist (e.g., Wever et al., 2014), we choose the simpler and more computationally efficient bucket scheme because its performance is not substantially different from others (Verjans et al., 2019) and allows us to use SNOWPACK with a setup that has been previously evaluated (Thompson-



Munson et al., 2023). This work uses the same model parameters as historical simulations performed over Greenland from
1980–2020 (Thompson-Munson et al., 2023).

## 2.2 Firn model atmospheric forcing

We use atmospheric reanalysis data from the Modern-Era Retrospective Analysis for Research and Applications,
version 2 (MERRA-2, Gelaro et al., 2017) as the foundation to force our idealized SNOWPACK firn model experiments.
MERRA-2 (available 1980–present) is a global, gridded reanalysis product with a horizontal resolution of 0.5° latitude by
0.625° longitude. We focus this study on the Greenland Ice Sheet and use only grid cells with a MERRA-2 ice coverage of at
least 50 %.

The SNOWPACK firn model requires several atmospheric variables as input and uses the MeteoIO library (Bavay
and Egger, 2014) to prepare the MERRA-2 forcing data. Since SNOWPACK requires input at half-hourly timesteps, the hourly
MERRA-2 data are converted to a higher frequency using nearest-neighbor and linear interpolations. MeteoIO reads in 15
MERRA-2 variables (Global Modeling and Assimilation Office (GMAO), 2015d, b, c, a) and, for each grid cell, outputs a
time series of air temperature, humidity, zonal wind, meridional wind, incoming shortwave radiation, incoming longwave
radiation, precipitation, and the precipitation phase. We use these prepared atmospheric forcing data as the basis for generating
the synthetic forcings.

## 2.3 Firn model experiments

For our idealized experiments, we require a realistic but constant climate forcing with a complete seasonal and diurnal
cycle, but no interannual variability nor trends. To generate this synthetic forcing, we select hourly MERRA-2 data from a
one-year period spanning 1 January 1991 through 31 December 1991 (Fig. A1). We choose this period because the net change
in firn air content over this year-long period is negligible (Thompson-Munson et al., 2023), which limits the potential
introduction of significant trends in the atmospheric data to the synthetic forcing. We verified that this choice of the model
year has a negligible impact on the control climate and on the idealized atmospheric warming and cooling experiments (not
shown).

To spin up SNOWPACK, we force the model with this synthetic constant climate (MERRA-2 data from the year
1991) until at least 150 m of snow, firn, and ice has accumulated or the bottom 50 m of the simulated firn column is solid ice
(density = 910 kg m$^{-3}$) with a firn depth of at least 10 m. This 50-m threshold is greater than in the similar spin-up procedures
used in Banwell et al. (2023) and Thompson-Munson et al. (2023) in order to achieve a baseline snow, firn, and ice column in
equilibrium that lacks any long-term trends. This procedure also filters out model grid cells that do not meet our chosen ice
thickness and firn depth thresholds, which removes some ablation zone grid cells and leaves us with 1724 locations for running
temperature experiments.

After the model spin-up, we run SNOWPACK for 100 yr with the same constant climate of the year 1991 repeated
100 times. This initial century represents the control period. We then assess the influence of idealized temperature changes on



the firn by running the model for an additional 100 yr with a stepwise air temperature perturbation included. For a warming experiment, we increase air temperature by 1 °C at $t = 100$ yr. In a cooling experiment, we decrease air temperature by 1 °C at $t = 100$ yr. As such, each of the 1724 grid cells has a 100-yr control period preceding 100-yr warming and cooling periods.

## 3 Results

**3.1 Response of Greenland firn air content to idealized atmospheric warming and cooling**

We first present the modeled ice sheet and climate mean state during the control period prior to any atmospheric warming or cooling (Fig. 1). Overall, the ice sheet and climate properties during this control period resemble the modern Greenland Ice Sheet. The control air temperature and snowfall are both generated from MERRA-2 and reflect the geometry and topography of the ice sheet (Fig. 1a, b). Consistent with observations and other climate models (e.g., Hanna et al., 2021),

the highest temperatures are along the low-elevation margins, and the lowest temperatures are found in the high-elevation ice sheet interior (Fig. 1a). The warmest areas within the model domain are along the steep southeastern margin, which is also where orographic precipitation causes the highest snowfall rates on the ice sheet (Fig. 1b). Some snowfall occurs in the south and along the western margin, but rates are very low in Greenland's interior and in much of the north. While the temperature and snowfall are derived from MERRA-2 and used to force the SNOWPACK firn model, the melt and firn air content are

calculated by the model (Fig. 1c, d). During the control period, melt occurs where the temperatures are highest. The most melt occurs in the southeast, and little to no melt is found in the ice sheet interior (Fig. 1c). Spatial variations in firn air content reflect the combined effect of snowfall and melt (Fig. 1d). The mean of 19.9±8.3 m and spatially integrated firn air content of 35621 km$^3$ are consistent with results from a previous SNOWPACK model study (Thompson-Munson et al., 2023). In the present study's control period, firn air content ranges from 0.2 to 40.4 m. The highest values occur where snowfall is high but

melt is low, and the lowest values are found along the margins in warm, high-melt areas (Fig. 1).

We next examine the changes in firn air content in response to idealized atmospheric warming and cooling. When perturbed from the control state (Fig. 1), idealized warming causes depletion of firn air content (Fig. 2a) while idealized cooling causes generation of firn air content (Fig. 2b). The largest changes in both warming and cooling experiments occur along the margins of the ice sheet but slightly inland from the ice edge. Warming and cooling also both lead to substantial change in the

southwest where melt rates are high, and in the southeast where snowfall is high (Fig. 1). While the firn air content response to warming and cooling by 1 °C are opposite, they are not equal in magnitude (Fig. 2c). At the end of the experiments, the spatially integrated firn air content changes by −9.7 % (−3441 km$^3$) in the warming scenario and by +8.3 % (+2954 km$^3$) in the cooling scenario. The standard deviation of post-perturbation spatially integrated firn air content is also greater in warming (891 km$^3$) than in cooling (769 km$^3$).

In addition to responses integrated over the entire ice sheet, we also contrast the firn response where melt exists ("wet firn") and where it does not exist ("dry firn"). The majority of firn air content change in both experiments occurs in wet firn rather than dry firn (Fig. 2d). Of the 3441 km$^3$ of air volume lost to warming, 78 % (2669 km$^3$) of the change occurs in wet



firn and 22 % (771 km$^3$) occurs in dry firn. For the 2437 km$^3$ gained in cooling, 82 % (2437 km$^3$) of the change occurs in wet firn and 18 % (518 km$^3$) occurs in dry firn. Idealized warming causes a greater change in both wet and dry firn, and the change in wet firn is greater than the change in dry firn for both warming and cooling.

To explore the timescales of firn air content changes, we contrast the rates of firn air content changes in response to atmospheric warming and cooling. In both experiments, firn air content rapidly changes in the years immediately following the temperature perturbation, and then the rate of change decreases with time as an equilibrium is slowly approached but not fully met within the 100-yr experiment (Fig. 2c). We explore the spatial dependence on the rate of change by calculating the time elapsed until a given percent change is reached following the warming or cooling perturbation (Fig. 3). For all three values investigated (±2.5 %, ±5 %, ±10 %), more grid cells achieve the given percent change within the 100 years following the perturbation in warming (Fig. 3a–c) compared to cooling (Fig. 3d–f). Following warming, firn air content changes by at least 2.5 % in 97 % of the grid cells compared to 73 % due to cooling. For warming and cooling, 69 % and 51 %, respectively, of the locations change by at least 5 % within 100 years. Finally, only 36 % of the grid cells in the warming scenarios and 28 % in cooling reach a 10% change in firn air content by the end of the simulations. Therefore, the firn air content response to warming is faster than its response to cooling. In both perturbation scenarios, the time required for firn air content to reach a given percent change is greatest in the interior (Fig. 3). The margins respond more quickly, with most grid cells reaching a ±2.5 %, ±5 %, or ±10 % change within a few decades. Notably, the southwest and northeast have some of the shortest elapsed times until a percent change is reached, and most grid cells in those areas change by at least ±10 % within 100 years. Although Greenland's firn doesn't appear to reach equilibrium in our experiments, we can still infer several important points about the response time. First, the response time depends on the sign of the climate change and is shorter for warming and longer for cooling. Second, the response time depends on the mean state climate, generally being longer at colder and drier sites than at warmer and wetter sites.

**3.2 Drivers of Greenland firn air content changes**

To understand the drivers of firn's response to idealized atmospheric warming and cooling, we examine firn changes as functions of control period summer air temperature and melt (Fig. 4). We first present the response of the colder (summer air temperatures < −4 °C) and drier (melt < 50 mm w.e. yr$^{-1}$) firn where the control period firn air content tends to be high (Fig. 4c, d). In these relatively cold and dry locations, the magnitude of the firn air content response to both warming and cooling is small (< 5 m) but increases approximately linearly as summer air temperature increases (Fig. 4a). In contrast, the very low melt rates appear to have very little influence on the magnitude of the response (Fig. 4b). The effects of atmospheric warming and cooling are opposite, and their strengths are nearly equal, though the effect of warming is persistently slightly larger in magnitude (Fig. 4c, d). While this colder and drier firn has a relatively small response governed primarily by summer air temperature alone, this is not the case for the warmer and wetter areas (Fig. 4 a, b). For regions with summer air temperatures between about −4 °C and 1 °C, and with melt rates between about 50 mm w.e. yr$^{-1}$ and 600 mm w.e. yr$^{-1}$, firn air content changes are very large (up to 16 m). Unlike in cold and dry firn, warm and wet firn responses to warming and cooling are not





approximately equal in magnitude. Rather, the warming effect is greater where firn air content initially exists (≥1 m) in the control period, and the cooling response is greater where the control period firn air content is close to 0 m. At the very warmest and wettest locations (e.g., summer temperatures >1 °C and melt rates >600 mm w.e. yr$^{-1}$), very little change in firn air content occurs in either warming or cooling, as very little air content initially exists (<1 m).

185    The similar effect of summer air temperature and melt on firn air content demonstrate the important connection between these two surface variables. Within the control period, areas with high summer air temperature have exponentially more melt (Fig. 5a). An additional nonlinear relationship exists between the control mean summer air temperature and mean temperature of the firn layer. Firn temperature increases approximately linearly with the air temperature in colder grid cells where melt is nonexistent or very small (Fig. 5b). However, when the air temperature exceeds ~−4 °C, there is dramatic change 190    in the relationship between air temperature and firn temperature.

Unlike the air temperature and melt, which are both altered in the perturbation experiments either directly or via the model's surface energy balance, snowfall and rainfall are unchanged in our experimental design. Thus, as expected, only weak relationships exist between firn air content changes and these precipitation variables (Fig. 6). At the lowest snowfall rates (below ~120 mm w.e. yr$^{-1}$) very little firn air content change occurs due to either warming or cooling (Fig. 6a). The greatest 195    changes tend to be concentrated in grid cells that experience moderate to high snowfall rates (~120 to ~1000 mm w.e. yr$^{-1}$), though some large changes occur at the highest snowfall rates. The relative strengths of warming and cooling do not appear to vary strongly with rainfall either (Fig. 6c). Rainfall shows a slightly stronger relationship to changes in firn air content, though substantial spread still exists (Fig. 6b). At the highest rainfall rates, the firn air content response is very small with changes less than 3 m in magnitude.

200  **3.3 Characterizing the diversity of local responses to temperature change**

Having provided an ice-sheet wide assessment, we next provide examples of firn air content responses from six locations with distinct mean and final states (Fig. 7). The six examples highlight the large range of summer air temperatures and melt rates as well as their impact on firn air content. They also demonstrate the complex interplay of the physical processes controlling firn in SNOWPACK. The coldest of these locations experiences no melt in the control, cooling, or warming 205    conditions and therefore represents a completely dry firn response (Fig. 7a). Atmospheric warming leads to a 14 % decrease in the control firn air content of 16.38 m and cooling leads to an 8 % increase by the end of the simulation. This greater response to warming is typical for the dry ice-sheet interior, and many grid cells exhibit a similar evolution of firn air content due to warming and cooling. In the next example, the control and cooling experiments produce no melt, but the warming experiment leads to a small amount of melt that transitions the firn in this location from dry to wet (Fig. 7b). The control firn air content 210    of 28.12 m decreases by 4 % due to warming and increases by 2 % due to cooling. Figure 7c depicts firn air content responses to the opposite wet/dry situation in which the control period and warming experiment experience melt but the cooling experiment decreases the air temperatures enough to eliminate melt and convert the wet firn to dry firn. In this case, the control firn air content of 15.74 m changes more due to cooling (+14 %) compared to warming (−11 %).



The final three examples are from warmer locations that experience melt in the control, warming, and cooling experiments (Fig. 7d–f). Where the control firn air content is 20.28 m and the mean summer air temperatures in all three experiments are below 0 °C, atmospheric warming leads to a much stronger firn air content change (−17 %) compared to cooling (+9 %) (Fig. 7d). Figure 7e shows firn air content results from a similar location on the ice sheet as in Fig. 7d, but here the control melt is more than twice as large. This results in a lower control firn air content of 7.34 m that is almost fully depleted due to warming (−96 %) but more than doubles due to cooling (+131 %). This is an example of a firn air content-limited response that occurs along many of the margins where the control firn air content is low enough that warming fully depletes it. The final example is from a location where the control mean summer air temperature is above the melting point and there is almost no firn air content to begin with (0.90 m) (Fig. 7f). The relative changes in firn air content appear to be large (−34 % from warming and +214 % from cooling), but the absolute changes are very small as warming almost fully depletes the firn air content and cooling increases it by a few meters.

To explore the complexities that meltwater introduces to the structure and properties of the firn, we examine the subsurface firn properties for the 10 years before and after the temperature perturbation for the two wettest and warmest local examples in Fig. 7 (Fig. 7e, f). We first explore the simpler of the two cases in which the effects of atmospheric warming and cooling are more easily explained. Here, control state summer air temperatures (−2.8 °C) and melt rates (367.1 mm w.e. yr$^{-1}$) are low enough to enable comparatively high firn air content (7.34 m) before perturbations are applied (Fig. 7e). In the upper 30 m of the firn column, the control density is less than ice density (910 kg m$^{−3}$ in the model) and has a strong seasonal signal that is preserved in buried layers over time (Fig. 8a, d). Following cooling, the density decreases and the denser control period layers are buried with time (Fig. 8a). The firn is also colder throughout the year (Fig. 8b), and the small amount of liquid water generated in the summer in the upper few meters decreases (Fig. 8c). Warming leads to the opposite responses with density (Fig. 8d) and temperature (Fig. 8e) increasing, and liquid water content showing a slight increase and then minimal change (Fig. 8f). In this case, the firn temperatures in the summer reach the melting point and the firn layer turns to ice with a thin layer of snow on the surface.

We also examine the subsurface properties for a more complex location where almost no firn air content initially exists (0.9 m) and the mean annual summer air temperature is above the melting point (3.1 °C) before the temperature perturbation (Fig. 7f). Here in the ablation zone, the upper 30 m is almost entirely ice with a layer of snowfall that almost completely melts away in the summer (Fig. 8g, j). When the air temperature decreases, the density decreases with time as new snow is not entirely melted away (Fig. 8g). This new addition of porous material creates open pore space for meltwater to be stored rather than run off the surface. As such, the liquid water content in the firn actually increases due to cooling (Fig. 8i). Water within the firn layer can remain a liquid at 0 °C or refreeze and release latent heat, which means that the firn temperature increases due to cooling (Fig. 8h). Since so little firn air content initially exists and the air temperatures are already above 0 °C, the warming scenario creates very little change in the firn properties (Fig. 8 j–l).

Having explored the detailed responses of firn in a few examples, we now characterize the relative strength of the atmospheric warming and cooling effects for each modeled location on the ice sheet. We use the mean firn air content in the





control period and the magnitudes of firn air content change to categorize each location into one of four "response relationships" (Fig. 9). In an equal response relationship, the difference in magnitudes of firn air content change due to cooling

and warming is < 5 %. Most grid cells fall into this category with 1282 of the 1724 grid cells having almost equal gain of firn air content due to cooling as the loss due to warming (Fig. 9c). It is worth noting that even in these cases where the responses to warming and cooling are very near equal, the response to warming is slightly but persistently greater in magnitude (Fig. 9b, d). A significantly greater warming response relationship describes 235 grid cells and occurs when the effect of warming outweighs that of cooling by at least 5 %. In comparison, a greater cooling response relationship describes just 22 grid cells.

The final category representing 185 grid cells is a firn air content-limited response, in which either the control firn air content is less than 1 m or warming depletes it to less than 1 m. Since firn air content cannot be negative, the response to warming has a lower bound of 0 m but the response to cooling is unlimited (Fig. 9a), meaning that these air content-limited responses tend to occur where the mean firn air content in the control period is very low (Fig. 9b). Categorizing each experiment pair into these four bins allows us to discuss spatial patterns in response strengths (Fig. 9c). The margins of the ice sheet are

characterized by either a greater response to warming, a greater response to cooling, or a firn air content-limited response. For the most part, the typically dry ice sheet interior has an equal response relationship. Although these locations are defined by a near-equal response, the vast majority of these grid cells have a slightly greater response to warming than to cooling when comparing absolute magnitudes of the responses (Fig. 9d).

### 3.4 Temperature–firn interactions responsible for firn air content changes

265         Having described the variety of firn responses to atmospheric warming and cooling, we now summarize the major pathways through which air temperature can alter firn air content. Figure 10 highlights many of these surface and subsurface processes that act together to produce the wide array of firn responses seen in our warming and cooling experiments. Changes in the compaction rate are the simplest and primary mechanism of changing firn air content in dry firn (Fig. 10). When the air warms and causes the firn to warm, the compaction rate, which is nonlinearly related to temperature, increases. This response

means that the firn compresses faster and firn air content ultimately decreases. In the case of decreasing the air temperature, the firn cools and slows the compaction rate, and the firn air content is higher than in the initial state. Additionally, the amount of air within the firn impacts the bulk thermal conductivity, allowing faster heat transport to deeper layers in the case of warming, and slower transport due to cooling.

         While the temperature–compaction relationship is the only major mechanism for firn air content in dry firn to change,

the introduction of meltwater adds complexity to the system. Wet firn's air content is dependent on the changing compaction rate, but also on processes related to surface melt and meltwater percolation (Fig. 10). Increasing the air temperature causes melting through a nonlinear relationship identified both in this work (Fig. 5) and others (e.g., Trusel et al., 2018). The process of melting removes porous material and fills pore space with meltwater, thus reducing firn air content. However, complexities arise from the presence and interactions of meltwater within the firn. When the liquid water replaces air, it increases the thermal

conductivity and the bulk temperature of the firn. If the water refreezes, it releases latent heat that further increases the firn



temperature and compaction rate (Pfeffer and Humphrey, 1996). For refreezing to occur, cold content (i.e., the energy required to bring firn to 0 °C) must be available (Vandecrux et al., 2020). Increasing the firn temperature reduces the cold content, which could limit refreezing and therefore the release of latent heat. However, cold content may be drawn from deeper firn via thermal conduction, so this negative feedback that buffers the effect of latent heat is likely minimal (Vandecrux et al., 2020).

Opposite to warming, decreasing the air temperature leads to less melt, which reduces the loss of porous material and reduces the amount of meltwater entering pore space (Fig. 10). With less meltwater occupying pore space, there is less liquid available for refreezing. As such, less latent heat is released. However, the availability of cold content in colder firn may allow for more refreezing and latent heat release that then warms the firn and enhances compaction. Whether firn temperature is ultimately decreased from advection or increased from amplified refreezing is dependent on the initial state of the firn and
leads to the variety of responses across the ice sheet (e.g., Fig. 9). The example of firn property changes due to cooling in Figs. 7f and 8g–i is reflective of these complex subsurface interactions. In the control simulation for this wet firn grid cell, most melt becomes runoff since so little pore space exists. Cooling the air temperature allows for less loss of porous material and thus more meltwater storage capacity. The increased meltwater within the firn increases the thermal conductivity, firn temperature, and latent heat of refreezing. As such, this is an example of atmospheric cooling leading to an initial increase in
firn air content that then results in a warmer firn layer due to meltwater storage.

## 4 Discussion

The key finding of this work is that the loss of firn air content due to atmospheric warming is greater than the gain in firn air content due to cooling in most regions and when integrated across the Greenland Ice Sheet. We identify this asymmetry both in wet and dry firn, and we attribute it to (1) the nonlinear relationship between firn temperature and compaction rate, (2)
the nonlinear relationship between air temperature and melt, (3) the additional warming of firn from the latent heat of refreezing, and (4) the dependency of the thermal conductivity on the material composition. Many of these processes have been found to be important to the response times of firn to climate change (Kuipers Munneke et al., 2015). The interplay of these relationships drives the variability of responses modeled across the ice sheet. Previous studies have examined the effects of idealized forcing perturbations on firn in theoretical frameworks (Kuipers Munneke et al., 2015; Meyer and Hewitt, 2017)
or with a few example locations (Arthern and Wingham, 1998; Li and Zwally, 2015). Our work expands on these studies to demonstrate the range of firn responses across Greenland, which depend on mean state climate variables like summer air temperature and melt. Despite the variability in responses, we find that on average Greenland's firn air content is more sensitive to warming than to cooling in most climate conditions and in both wet and dry areas.

An additional and important contributor to the asymmetric firn response is the immediate buffering effect of cooling
in very warm and wet areas. Storage of meltwater in newly created pore space reduces surface runoff but does not provide a long-term buffer. In a cooling climate, this mechanism may slow or stall the growth of the ice sheet's meltwater retention capacity. On the other hand, loss of pore space through atmospheric warming does not face a similar obstacle. The firn air



content response may be limited by the amount of initial air available, but where loss of pore space is not possible, mass loss occurs and warming still affects runoff. While this negative feedback dampens the effect of cooling, no such limitation exists

for warming. This is demonstrated through the rapid expansion of the ablation zone responsible for recent surface runoff in Greenland (Noël et al., 2019; Tedstone and Machguth, 2022). Our findings similarly demonstrate the rapid effects from warming, but the asymmetric firn response implies that the reversal of this process in a cooling climate would occur more slowly.

More broadly, our findings provide an additional mechanism that contributes to the asymmetric growth and decay of

ice sheets. Fyke et al. (2018) describes several processes that play a role in this asymmetry, such as the melt–elevation and melt–albedo positive feedback loops that can enhance ice sheet mass loss. These two feedbacks primarily relate to the ablation zone and lower-elevation areas experiencing melt. However, we present a mechanism that acts in the accumulation zone. Since wet firn and dry firn processes are both nonlinearly related to temperature, the asymmetric temperature–firn response operates in all areas where firn exists. Although the asymmetric response describes firn air content changes rather than mass changes,

the two are closely related through the firn's meltwater retention capability (Harper et al., 2012). Further, these nonlinearities are largest at the edges of the accumulation zone, that is the regions that can transition from accumulation zone into ablation zone, or back. These processes suggest that the ablation zone can grow in area more quickly than it can shrink. Thus, we describe an asymmetry in the behavior of Greenland's firn that may contribute to the ice sheet's overall asymmetric mass changes.

This work also provides important insight into how firn evolves not only in our modern warming climate, but in a cooling climate as well. Much research has been conducted on the signature of recent atmospheric warming on the firn layer, but fewer opportunities exist to observe the effects of cooling. Though not directly assessing decreasing air temperatures, Rennermalm et al. (2022) found that recently depleted pore space can be temporarily recovered with several years of average or below-average melt. Our work shows similar generation of pore space as cooling reduces melt and compaction rates. In a

cooling climate, we expect firn to respond more slowly and to a lesser degree when compared to modern observations of firn air content depletion. As such, in the absence of changes to other climate variables, a greater degree of cooling is required to achieve the same strength firn response in warming. This suggests that reversing the effects of modern climate change through air temperature change alone would require disproportionately more cooling.

The idealized design of the study provides important insights into firn–climate interactions, but like all idealized

designs has strengths and limitations that are important to acknowledge. The universally applied stepwise perturbation itself is simple to implement and interpret. In addition to being simple, this step-change experimental design enables direct quantification of response timescale. The inspiration for this step-function design comes from widely used climate sensitivity experiments that assess climate response to instantaneously changing carbon dioxide concentrations (e.g., Manabe and Wetherald, 1975; Sherwood et al., 2020) as well as firn modeling experiments quantifying the response times of idealized firn

to atmospheric perturbations (Kuipers Munneke et al., 2015). While a lot can be learned, we also are fully aware of limitations of our idealized experiments. Although the idealized climate we apply is based on observations, it does not represent a realistic



200-yr climate. Temperature change is highly unlikely to occur equally across the entire ice sheet in the span of a single timestep (30 min) and remain unchanged for a century. We only examine the effects of changing a single forcing variable, but several atmospheric properties are likely to vary in realistic climate change scenarios. We choose air temperature because of

its strong impact on both dry and wet firn properties, and its relationship with surface melt and runoff that have driven recent mass loss (van Angelen et al., 2013). Additionally, we use firn air content because it is indicative of the meltwater storage capacity of the ice sheet. However, not all air-filled pore space is available for storing meltwater since access may be blocked by low-permeability ice slabs that seal off deeper, porous firn (MacFerrin et al., 2019). In this work, we do not distinguish between accessible and inaccessible firn air content. Finally, we only consider how temperature perturbations alter two key

firn processes: compaction and melt. However, we note that other firn processes are altered by a change in air temperature. For example, we do not explore the effects of sublimation because it is tied to both humidity and wind speed in addition to temperature (Lenaerts et al., 2019). Still, we acknowledge that surface mass balance processes like sublimation affect firn air content and are altered by the prescribed temperature perturbations.

**Conclusion**

360     This study uses idealized warming and cooling experiments to characterize the response of simulated firn to air temperature change. We rely on a physics-based firn model run with an idealized forcing, which allows us to isolate the effect of temperature perturbations of equal magnitude (1 °C) but opposite sign on firn air content. We find that Greenland firn air content loss from atmospheric warming is greater than the gain from cooling both locally and integrated across the ice sheet, and we attribute this asymmetry to the nonlinear effects of temperature on firn processes. The variable magnitudes of local firn

air content responses to warming and cooling result from the mean state climate as well as the complex interplay of temperature–firn interactions. Our results agree with modern observations of firn air content depletion from warming, but they offer new insight into firn air content changes in a cooling climate. By examining both atmospheric warming and cooling, we identify an important asymmetry in firn that likely contributes to the overall asymmetric mass changes of the Greenland Ice Sheet. Based on our interpretation of these findings, reversing the effects of recent warming on firn air content, melt, and

runoff would require a greater degree of cooling.



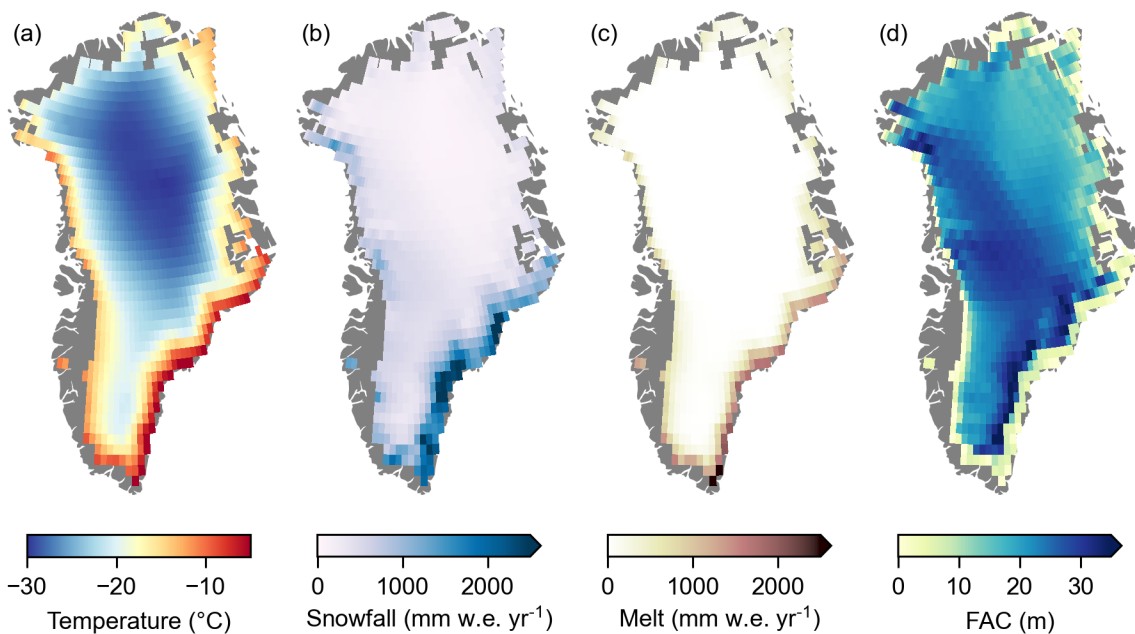

**Figure 1: Greenland Ice Sheet mean annual (a) temperature, (b) snowfall, (c) melt, and (d) firn air content (FAC) from the 100-yr control climate prior to the warming and cooling experiments.**





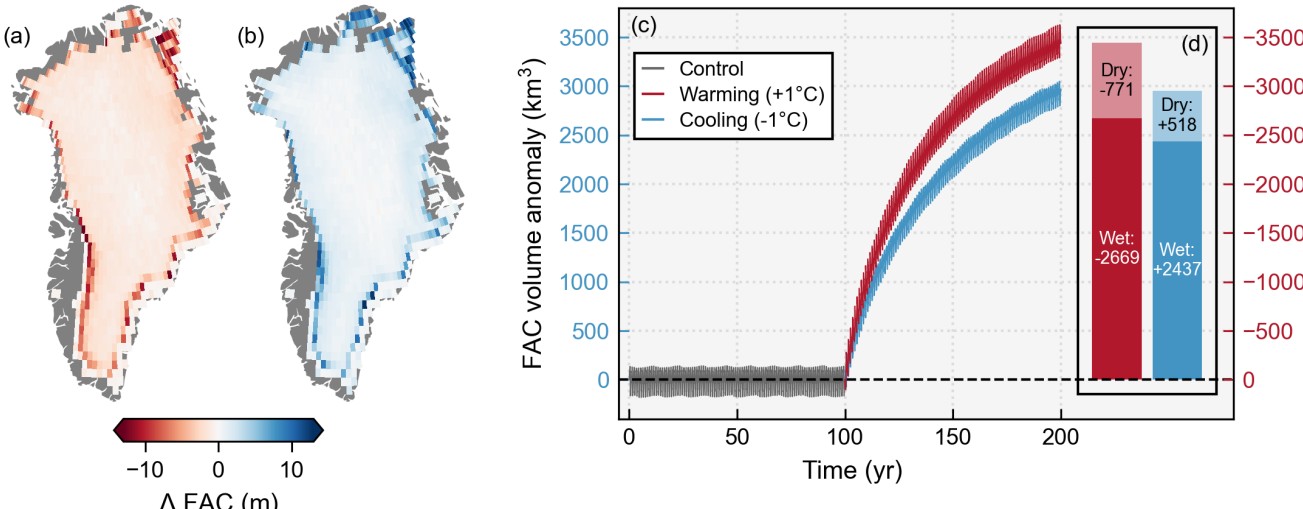

**Figure 2: Change in firn air content (FAC) for the Greenland Ice Sheet calculated as the final firn air content minus the mean of the**
**control conditions for (a) warming by 1 °C and (b) cooling by 1 °C. (c) Time series of the firn air content volume anomaly integrated**
**over the full ice sheet for the control (gray line), warming (red line), and cooling (blue line) periods. The control period and cooling**
**experiment use the positive values on the left _y_-axis and the warming experiment uses the negative values on the right values on the**
**right _y_-axis. (d) Changes in spatially integrated firn air content in km³ at the end of the experiments partitioned into wet firn (melt**
**is present) and dry firn (melt is not present) areas for warming (red) and cooling (blue).**





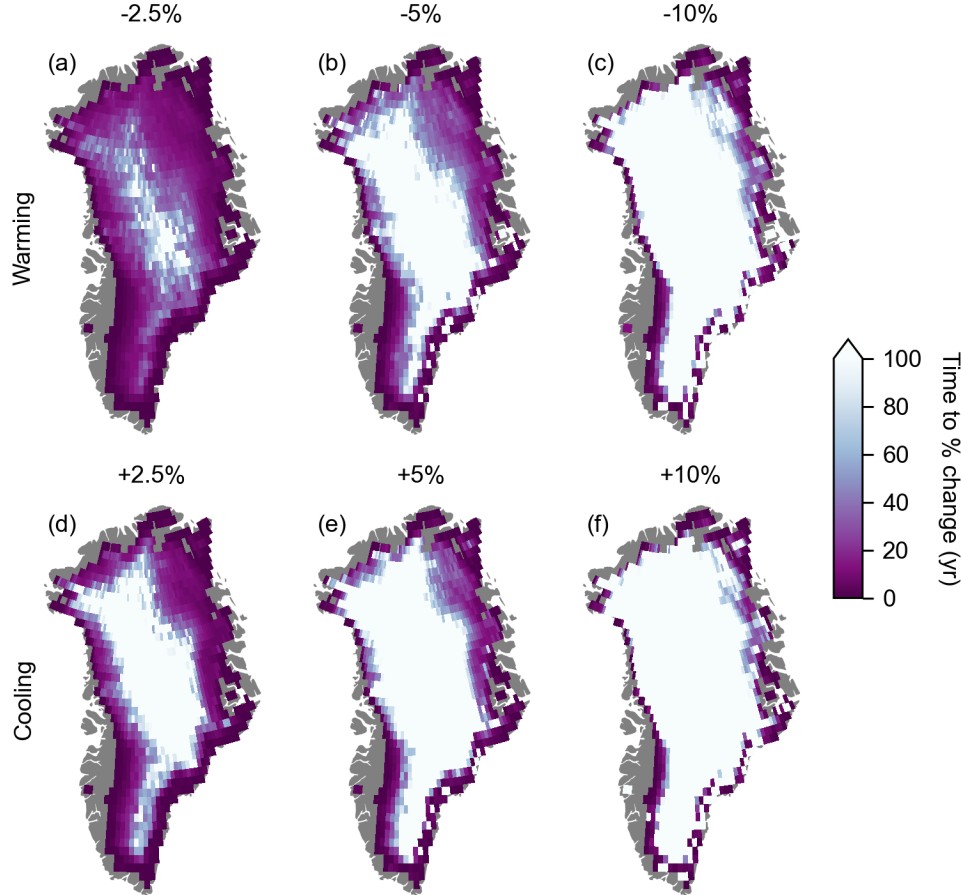


**Figure 3: Elapsed time to reach a given percent change (shown above each panel) in firn air content following the control mean. Results are shown from both (a–c) warming by 1 °C, and (d–f) cooling by 1 °C. The lightest shading indicates that the given percent change has not been achieved within the 100-yr warming or cooling experiment, and thus the elapsed time is >100 yr.**




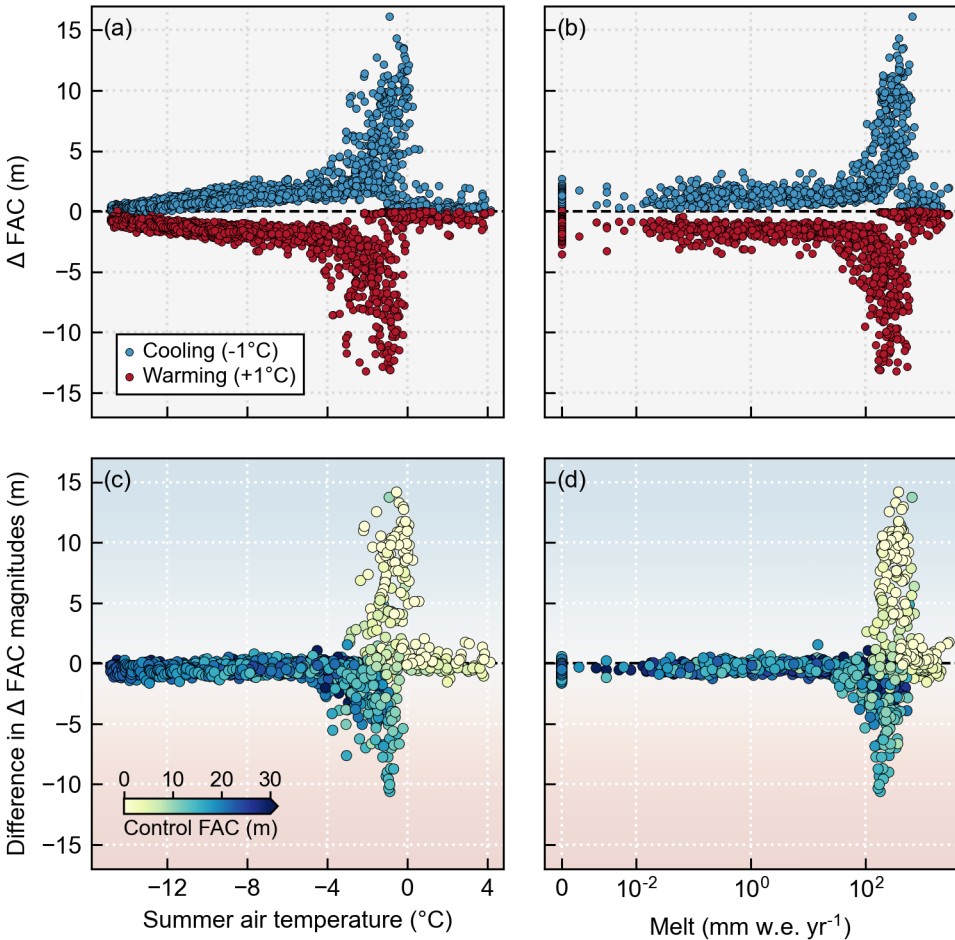

**Figure 4: (a–b)** The change in firn air content (FAC) for each grid cell calculated as the final value minus the control mean as a function of the control mean (a) summer air temperature and (b) melt. Blue markers indicate the effect of cooling and red indicate the effect of warming. **(c–d)** The difference in the magnitudes of firn air content changes due to warming and cooling. The difference is calculated as the magnitude of the cooling change minus the magnitude of the warming change, such that positive values indicate a greater response to cooling (blue background shading) and negative values indicate a greater response to warming (red background shading). The markers are colored by the control mean firn air content. Note the semi-logarithmic *x*-scales used for melt in (b) and (d).





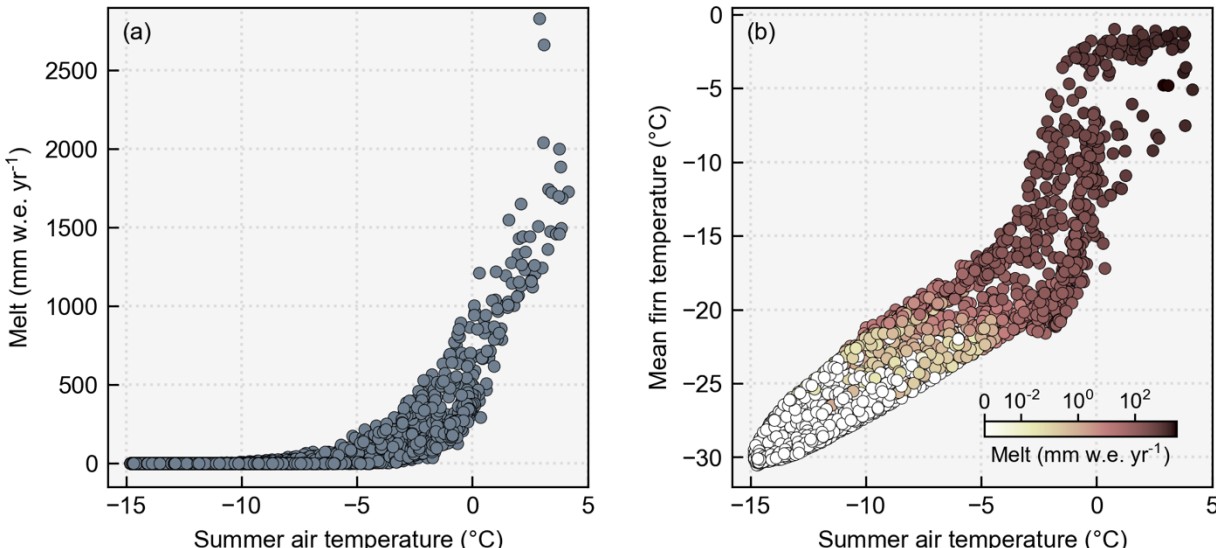

**Figure 5: Control mean (a) melt and (b) firn temperature as functions of the control mean summer air temperature for all modeled locations. The color of the points in (b) represents the control mean melt.**





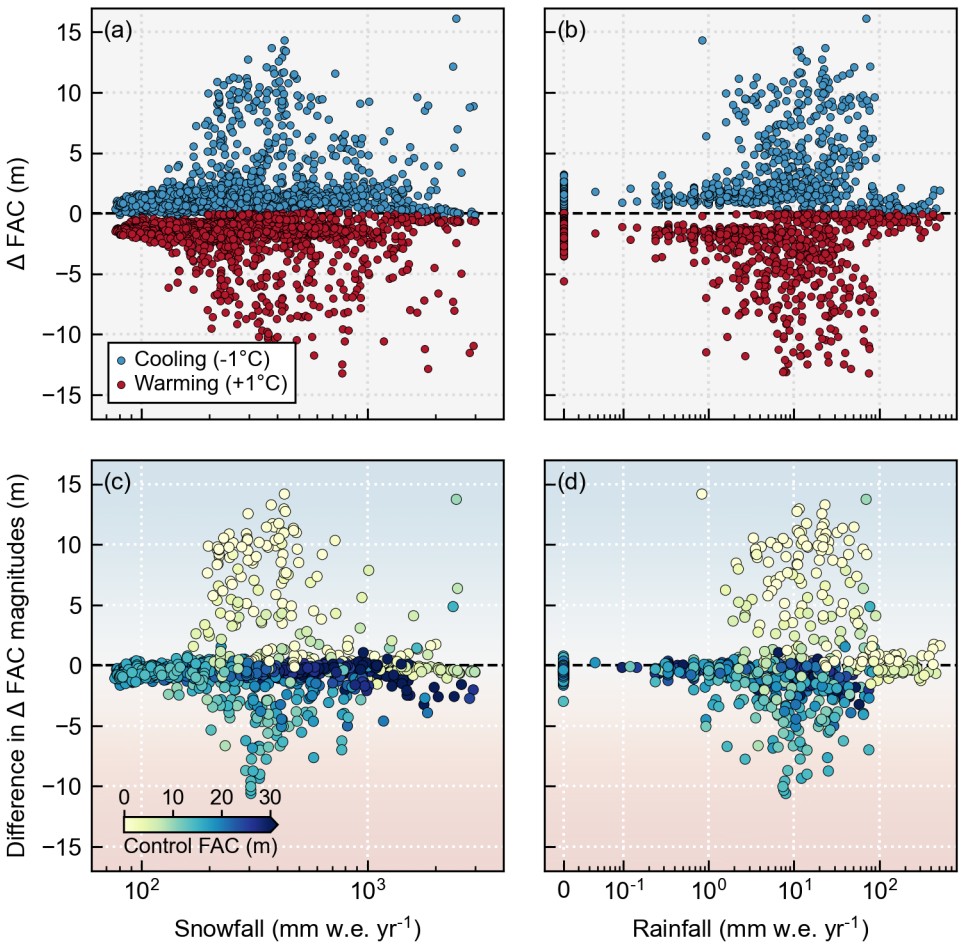

**Figure 6: (a–b) The change in firn air content (FAC) for each grid cell calculated as the final value minus the control mean as a function of the control mean (a) snowfall and (b) rainfall. Blue markers indicate the effect of cooling and red indicate the effect of warming. (c–d) The difference in the magnitudes of firn air content changes due to warming and cooling. The difference is calculated as the magnitude of the cooling change minus the magnitude of the warming change, such that positive values indicate a greater response to cooling (blue background shading) and negative values indicate a greater response to warming (red background shading). The markers are colored by the control mean firn air content. Note the semi-logarithmic $x$-scales used in all panels.**



**Table 1: Mean summer air temperature and annual melt in the control, cooling, and warming scenarios for the six examples shown in Fig. 7.**

| Corresponding figure panel | Control summer air temp. (°C) | Cooling summer air temp. (°C) | Warming summer air temp. (°C) | Control melt (mm w.e. yr⁻¹) | Cooling melt (mm w.e. yr⁻¹) | Warming melt (mm w.e. yr⁻¹) |
|---|---|---|---|---|---|---|
| Fig. 7a | −10.1 | −11.0 | −9.0 | 0.0 | 0.0 | 0.0 |
| Fig. 7b | −10.0 | −10.9 | −8.9 | 0.0 | 0.0 | 1.0 |
| Fig. 7c | −7.2 | −8.1 | −6.1 | 0.4 | 0.0 | 0.8 |
| Fig. 7d | −4.5 | −5.4 | −3.4 | 170.3 | 91.0 | 235.0 |
| Fig. 7e | −2.8 | −3.7 | −1.7 | 367.1 | 203.0 | 466.6 |
| Fig. 7f | 3.1 | 2.2 | 4.0 | 2663.0 | 2162.9 | 2999.6 |

415

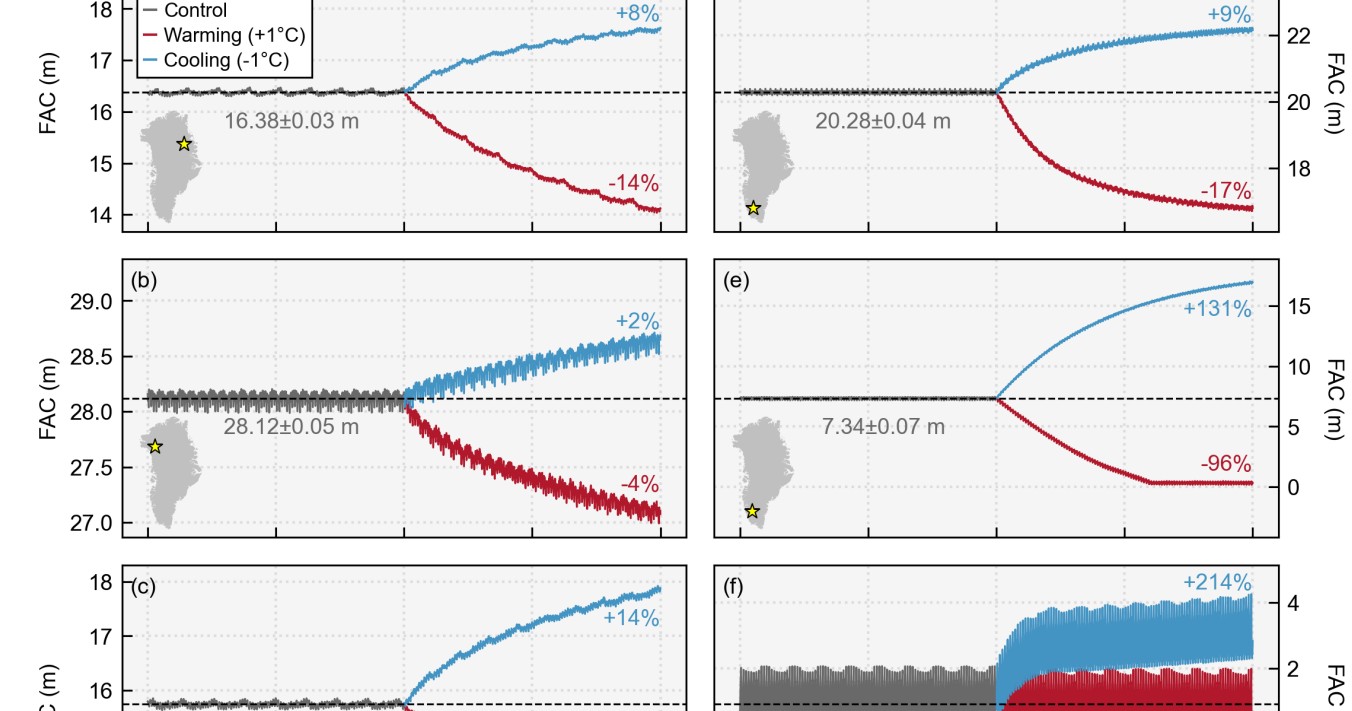

**Figure 7: Firn air content (FAC) over time in the control period (gray line), warming experiment (red line), and cooling experiment (blue line) from six locations shown in the inset maps as yellow stars. Numbers in gray show the mean and standard deviation of the firn air content in the control period, and red and blue percentages show the total percent change due to warming and cooling, respectively. Note the different *y*-axis scales.**

420



**Figure 8: Time–depth plots showing firn properties (density, firn temperature, and liquid water content) for the upper 30 m of the firn column for the 10 yr before and after the temperature perturbation (vertical black dashed line) for two example locations. (a–f) Results from a grid cell in southwest Greenland as seen in Fig. 7e. (g–l) Results from a grid cell in southeast Greenland as seen in Fig. 7f. The responses to cooling are shown in panels a–c– and g–i, and the responses to warming are shown in panels d–f and j–l. Inset maps in panels (f) and (l) show each example's location on the ice sheet as a yellow star.**



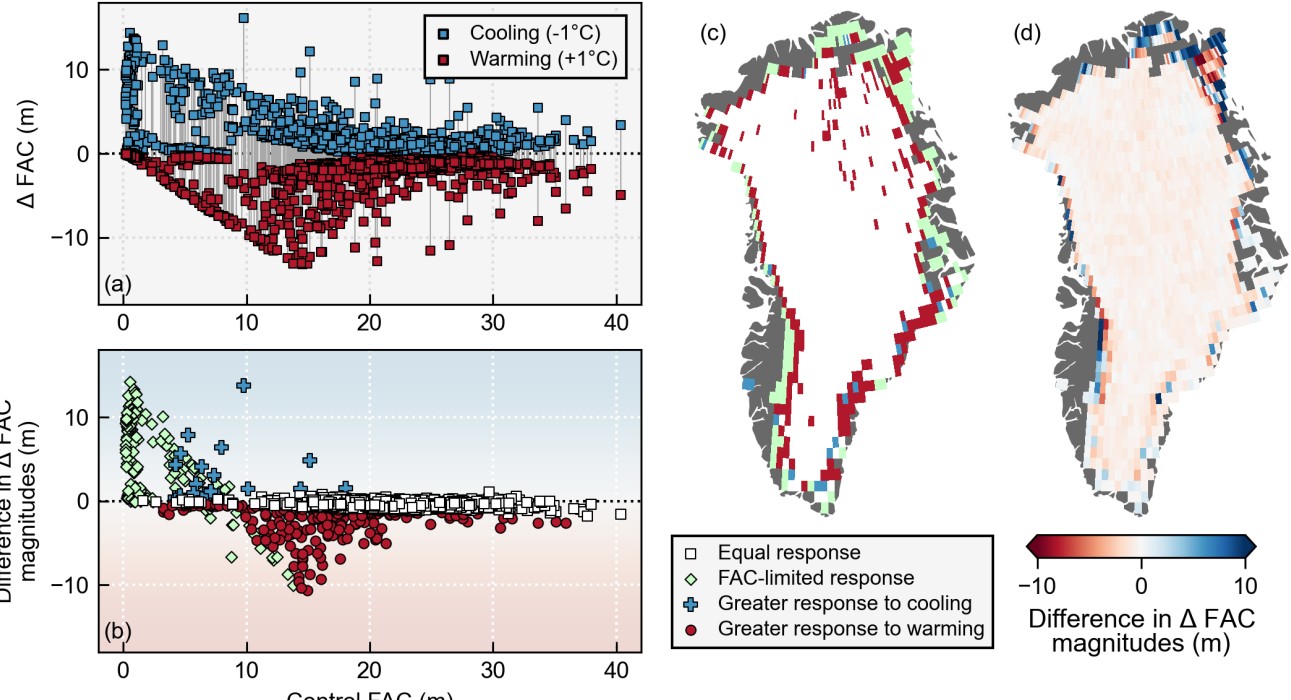

**Figure 9: Types of firn air content (FAC) response relationships across the ice sheet. (a) The change in firn air content (FAC) for each grid cell calculated as the final value minus the control mean as a function of the control mean firn air content. Blue markers indicate the effect of cooling and red indicate the effect of warming; gray vertical lines connect the warming and cooling responses from each grid cell. (b) The difference in the magnitudes of firn air content changes due to warming and cooling. The difference is calculated as the magnitude of the cooling change minus the magnitude of the warming change, such that positive values indicate a greater response to cooling (blue background shading) and negative values indicate a greater response to warming (red background shading). The different marker shapes and colors differentiate the four types of responses shown in the legend to the right. (c) Locations of the different response types across the ice sheet. (d) The difference in magnitudes of firn air content change due to cooling and warming shown in map view.**





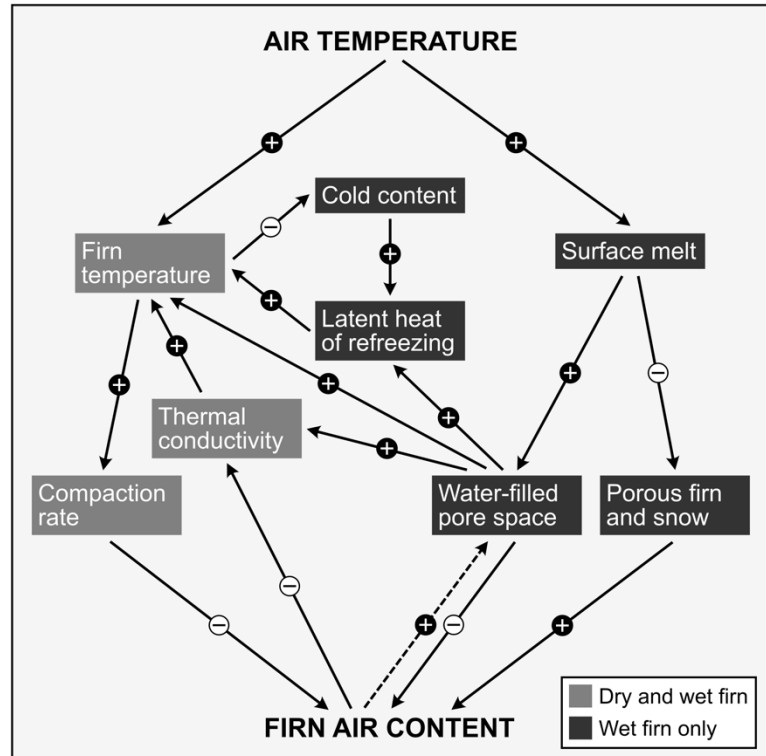

**Figure 10: Schematic of air temperature-driven interactions within the firn layer. Arrows with plus signs indicate a direct relationship while arrows with minus signs indicate an inverse relationship. The dashed arrow represents a relationship that only exists where runoff occurs and firn air content increases. Dark gray boxes show processes occurring in the presence of melt (i.e., in wet firn only) and light gray boxes show processes occurring in both dry and wet firn.**

445



## Appendix A

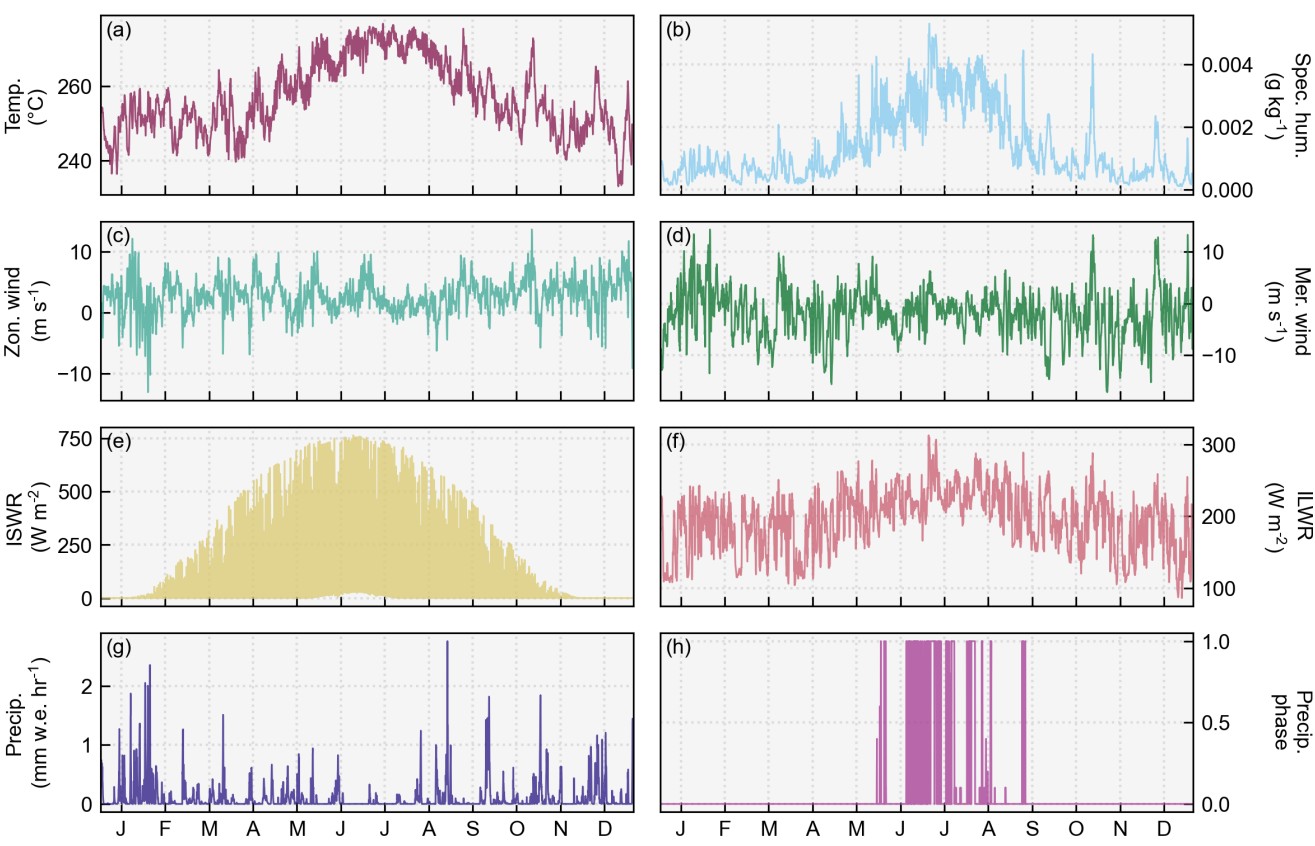

**Figure A1: Example of one year (1991) of hourly unperturbed atmospheric forcing data for a single location (69.5°N, 27.5°W). (a) Air temperature, (b) specific humidity, (c) zonal wind, (d) meridional wind, (e) incoming shortwave radiation, (f) incoming longwave radiation, (g) precipitation, and (h) precipitation phase. The precipitation phase is a fractional value ranging from 0 to 1, where 0 is all solid precipitation and 1 is all liquid precipitation.**



*Code and data availability.* The NASA GSFC MERRA-2 data are available at https://disc.gsfc.nasa.gov/. The code to run the SNOWPACK firn model is available at https://github.com/snowpack-model/snowpack. The code and data used to generate the figures for this manuscript are available on Zenodo at https://zenodo.org/doi/10.5281/zenodo.10069582. Due to their large size, the forcing dataset and raw model output are stored on the University of Colorado's PetaLibrary, which is not publicly available. As such, please contact the authors for access to the raw data. However, all code used to generate the forcing data, prepare the model output, and create the figures can be found on GitHub at https://github.com/MeganTM/greenland-firn-experiments.

*Author contributions.* MTM, JEK, and BRM all designed the study and contributed to the writing of the manuscript. MTM generated the forcing data, ran the model, and processed the output data.

*Competing interests.* The authors declare that they have no conflict of interest.

*Acknowledgments.* This research has been supported by the National Aeronautics and Space Administration (NASA) Earth Sciences Division (grant no. 80NSSC20K1727) and the Cooperative Institute for Research in Environmental Sciences (CIRES). The authors acknowledge Nander Wever for his help setting up SNOWPACK, and Colin Meyer for his helpful discussions. This work utilized the Alpine high performance computing resource at the University of Colorado Boulder. Alpine is jointly funded by the University of Colorado Boulder, the University of Colorado Anschutz, and Colorado State University. Data storage supported by the University of Colorado Boulder 'PetaLibrary.'



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
