# Peer review of "Greenland's firn responds more to warming than to cooling"

_EGUsphere, 2023_

## Author Comment (AC1)

Overview
==========

This is a modeling study to investigate the response of firn to changes in air temperature. The investigators run a simple step-forcing experiment to determine the spatial and temporal responses of a simulated firn column over Greenland, to step changes of 1 degree C in both warming and cooling. The experiment and procedures are well-described and the paper is generally well written. Many interesting insights are illustrated in the paper, and it looks to be an interesting contribution to the literature on firn in a warming world. While the main conclusions may be unsurprising to those who think about firn a great deal, the conclusions are well-founded and are well worth publishing. I have mainly a few relatively minor issues with the existing study and they should be straightforward to address.

We thank the referee for reviewing the manuscript and providing constructive suggestions and feedback. In particular, we are grateful for the suggestions that have made the manuscript clearer and more digestible to the reader. We have made modifications to the text and figures and believe the overall manuscript has been improved following the referee's feedback. Please find below our responses to comments in blue text, and specific changes made to the text in *blue, bold, italic font.* Line numbers refer to the original manuscript.

Minor issues
==============

- Timescale of the experiment: As the authors note (lines 151-152 and 165), the model in the perturbed state (both warming and cooling) comes _close_ to equilibrium (as shown in figures 2 and 7 for example) but doesn't quite get there. It would definitely be a more satisfying experiment to run these models to equilibrium. That would also allow an additional result- the quantification of the total change in FAC for the changes, and the total response time to equilibrium (bringing home the point the authors make about the dependence of response time on mean state climate- in lines 165-167).

We agree about the utility of experiments that run to equilibrium. We note that this has previously been done in Kuipers Munneke et al. (2015). As such, our study focuses on features of the firn that emerge even though the experiments don't quite reach equilibrium. Based on the results of this previous paper, we do not see evidence that running our experiments out longer will change our main results. For the ice-sheet-integrated response to reach equilibrium, each individual grid cell would have to reach an equilibrium as well. As shown in the original Figure 7, the time required to reach equilibrium varies in each example. This means that the timing for the ice-sheet-integrated response could be controlled by only a few grid cells. As such, investigating why certain cases reach an equilibrium faster than others would be more informative for understanding timescales. This has been done in Kuipers Munneke et al. (2015), so we have chosen not to repeat a similar set of experiments.

- Additional experiments: It seems that once the forcing framework is in place and the perturbations are set for 1 degree, it would not be difficult to 'turn the crank' on other perturbation experiments; most simple would be doubling the warming or cooling, but another more interesting question would be what is the magnitude of cooling required to match the change in FAC from the 1 degree warming experiment? There's an extent to which the answer to the existing experiment is self-evident to folks who study firn...additional experiments like this might make this a more widely-cited paper.

The primary goal of this paper is to compare warming and cooling responses using a highly idealized experimental set-up. While a key finding is that the warming effect on firn air content is greater than the cooling effect, the exact values of we report for firn air content change correspond to this ideal framework. While the exact values can change slightly, the main results do not depend on starting climate. Our main results are interesting, and also make physical sense. Thus, we focus on presenting this relative difference in firn air content change rather than the exact values or running experiments to answer questions like the reviewer raises here (i.e., "what is the magnitude of cooling required to match the change in FAC from the 1 degree warming experiment?"). While additional experiments would no doubt be interesting, we do not see evidence that they would change our primary science findings. On a practical note - the computational expense and storage requirements for running additional experiments are substantial given our resources so it is not so easy to 'turn the crank' as the reviewer suggests. While we agree that these subsequent ideas are interesting, they are substantial enough ideas that they could be left for additional studies.

- Appropriateness of title, given Figure 9c: at and near line 250, the authors state that the majority (1282/1724 = 74%) of the grid cells fall into the category labelled "Equal response" in figure 9. This is graphically shown in figure 9c. Given this statement, does the title (claiming unequal response) make sense? I certainly concede the point made in lines 252 and 262 that within this 5% bound the warming is usually greater (pinkish tint in Figure 9d). Figure 2c makes the point (of the title) well but then it's a bit undermined by the later discussion of figure 9c. Maybe instead of "Greenland's firn" in the title you might insert "Greenland's marginal firn"? Or change some of your binning here in this paragraph to make the point in the title more clearly made? Perhaps this is pedantic but seemed worth discussing.

As it is, the title conveys our most important result: both in the Greenland Ice Sheet interior and closer to the margins, the response to warming is greater. More importantly, we're communicating that when integrated across the full ice sheet, the warming response is greater. With that in mind, we have chosen to keep the title the same.

However, we appreciate the reviewer's point here about the title and the use of "equal response" within the paper. Even in the cases we originally labeled as "equal", the response to warming is slightly greater, but only by a small amount. To avoid using the word "equal" when the response magnitudes are not actually equal, we have instead changed how we describe these "small difference" relationships. We have changed the classification in the original Figure 9 (Figure 8 in the updated manuscript) to call this category "small difference (<5 %)" responses. We have also made changes to the text in Section 3.3 that describes this figure. Below are our modifications for sentences in lines 249 and 260, respectively.

*In a small difference response relationship, the difference in magnitudes of firn air content change due to cooling and warming is <5 %. Most grid cells fall into this category with 1282 of the 1724 grid cells having a similar gain of firn air content due to cooling as the loss due to warming (Fig. 8c). It is worth noting that even in these cases where the responses to warming and cooling differ by a small amount, the response to warming is slightly but persistently greater in magnitude (Fig. 8b, d).*

*For the most part, the typically dry ice sheet interior has a response relationship with small differences (<5 %) between warming and cooling. Although these locations are defined by a similar-magnitude response, the vast majority of these grid cells have a slightly greater response to warming than to cooling when comparing absolute magnitudes of the responses (Fig. 8d).*

The updated figure (originally Figure 9 but relabeled as Figure 8 in the revised manuscript) is shown below.

[Figure]

*Figure 8: Types of firn air content (FAC) response relationships across the ice sheet. (a) The change in firn air content (FAC) for each grid cell calculated as the final value minus the control mean as a function of the control mean firn air content. Blue markers indicate the effect of cooling and red indicate the effect of warming; gray vertical lines connect the warming and cooling responses from each grid cell. (b) The difference in the magnitudes of firn air content changes due to warming and cooling. The difference is calculated as the magnitude of the cooling change minus the magnitude of the warming change, such that positive values indicate a greater response to cooling (blue background shading) and negative values indicate a greater response to warming (red background shading). The different marker shapes and colors differentiate the four types of responses shown in the legend to the right. (c) Locations of the different response types across the ice sheet. (d) The difference in magnitudes of firn air content change due to cooling and warming shown in map view. The thin black line in (c) and (d) represents the ice sheet outline.*

Mainly small issues, line-by-line
=====================================

- Line 32: I would argue that firn air content and density are not really separate properties in this context; just different ways of describing the same property. Since air content is the key term used in the title and throughout, I'd either suggest a sentence relating the two, or leaving density out of it entirely.

This is a great point that another reviewer also brought up (see response to Referee #2, line 137). We have made the following changes to this paragraph:

"such as air content and density" → *like air content*
"Changes in firn air content and firn density" → *Changes in firn air content*

We kept one distinction between density and firn air content because we refer to specific papers that looked at one or the other variable.

- Line 33 Not sure MacFerrin et al 2022 is the right reference for the concept of compaction (they didn't come up with it!). Not clear a reference is needed here, but if so, please go back further in the literature- MacFerrin et al 2022 cite many other studies like Herron and Langway 1980, Morris and Wingham 2014, among others. If you want to stick with references already existing in the manuscript I'd suggest Herron and Langway 1980 and Arthern and Wingham 1998.

Thank you for pointing this out. Following your suggestion and those of another reviewer, we have removed "(MacFerrin et al., 2022)".

      - Line 80: non-modelers might not know what a "bucket scheme" is; suggest a short description.

We have added the following brief description to the end of that sentence:

      *in which liquid water is moved downward when a layer's water holding capacity is exceeded.*

      - Line 122: By "modern Greenland Ice Sheet" I think you mean 'holocene', right? It should really reflect
something like the 1991 Greenland Ice Sheet, which is definitely changed in the past three decades (thus "modern" does not mean 2023).

      This is a great point. We have removed this sentence since we feel it is imprecise and adds very little to this section. In response to this comment and another referee's we have specifically mentioned that we
mean the 1991 climate. Please see our response to Referee #2, line 84 for additional explanation.

      - Line 139: presumably the reason the big changes are slightly inland from the edge is that at the edge FAC is zero to begin with (ablation zone)? If so, this sentence may either not be needed or could include this point...

      This sentence has been changed to reflect suggestions from another referee. Please see our response to Referee #2, line 150 for additional explanation.

      - Figure 4: It took me a while to understand the observations made in lines 181-182, because there's a lot
going on in this figure. Perhaps in lines 181-182 refer to the colorbar for 4c,d to remind readers that the initial FAC in the control is indeed shown in those panels...

      Thank you for pointing this out. We have added a reference to Figure 4c, d in the text.

- Figure 6: if the editors want to save space, this would be a good figure to cut, as it illustrates essentially a null result which was expected.

      We appreciate this suggestion and have moved this figure to the appendix. Figure numbers have been updated to reflect this change and the main text now includes only 9 figures while the appendix has 2.
      - Line 224 and Figure 7f: Also a more general comment about this figure- the annual/seasonal cycles of changes in FAC are apparent particularly in 7f but the differences from the max in 7f and a min in 7e are pretty stark; this is worth discussing at least a little bit.

The first-order seasonality in the firn air content records is primarily a result of two surface mass balance components: snowfall and melt. Since we are not modifying the precipitation variable at all, the seasonality here is largely just a reflection of the melt, and therefore the air temperature. There are of course other processes like firn temperature changes modifying the compaction rate, which have a higher relative importance in the dry firn zones compared to in the wet firn zones. In an effort to not distract the
reader from our main points, we have chosen to leave out a discussion on the seasonality.

      - Line 239: If we are genuinely in the ablation zone, the snowfall cannot "almost completely" melt away in summer, it must completely melt away (and thus there would be by definition 0 FAC). Call this "here near the equilibrium line" instead of "here in the ablation zone".

      This is an excellent point and we appreciate this feedback. We have made the suggested change.

      - Line 250: The fact that the large majority of grid cells fall into the "almost equal" category sort of implies that the title of the paper might be an overgeneralization. See my 'minor point' above on this.

Thanks for the specific line reference here. We have addressed this comment above (see line 76 of this document).

- Line 272: instead of the equivocal "impacts the bulk thermal conductivity", state the generalized
relationship (ie increasing air content reduces conductivity)- it may be obvious to some readers but not all; it's implied in lines 272-3 but could be made more clear to the reader.

We appreciate this suggestion and have changed the sentence to:

*Additionally, increasing air content reduces the bulk thermal conductivity, allowing faster heat transport to deeper layers in the case of warming, and slower transport due to cooling.*

- Figure 10: This is a pretty complicated figure, which may be the point, but it's challenging to follow the description in the text along with the figure. One option would be to have letters or numerals on each box,
so that instead of referencing 'Figure 10' in the text, you can refer to a specific box or set of boxes in the figure.

We agree that this is a complex figure, and we have made the suggested changes so it is more easily understood. There are now letters labeling each box on the figure, and we have added references to the
specific boxes throughout the text in Section 4.1. The modified figure is shown below.

[Figure]

- All of section 3.4: It's not clear that these are actually results, and they read like discussion. Consider
putting them in section 4.

We have moved this section to the discussion and split the new discussion section into three subsections.

- Line 299: here you state "most regions" have a greater change in FAC from warming than from cooling-
this needs to be reconciled with the statements around Line 250 and in figure 9 where most of the area of the ice sheet falls into the "equal response" category. Also see my 'minor point' about the title of the paper above.

We thank the referee for noting this. We have removed "in most regions" so the sentence is better
supported by our findings.

- Line 345: "While a lot can be learned" a but colloquial. Suggest changing to "much can be learned".

Done.

- Line 547: reference to Kuipers Munneke et al 2015: Van den Brooke's name is broken up in an incorrect way, it should be Van den Brooke, M. R. (note I didn't check all of these references I just noticed this one, suggest double-checking the typesetting- probably a simple BiBTeX issue).

Thank you for finding this error. The reference manager software incorrectly split up some of the last names. We have fixed the noted issue and checked the other citations as well.

**References used in this response**

Kuipers Munneke, P., Ligtenberg, S. R. M., Suder, E. A., and Van den Broeke, M. R.: A model study of the response of dry and wet firn to climate change, Ann. Glaciol., 56, 1–8,
https://doi.org/10.3189/2015AoG70A994, 2015.

---

## Author Comment (AC2)

General

This is an interesting paper that investigates the response of the Greenland ice sheet firn layer to idealized positive and negative temperature perturbations on century time scales. The paper is well and concisely written, and the figures are of excellent quality. My comment are therefore relatively minor and should be fairly easy to address.

We thank the referee for their careful review of the manuscript and constructive feedback. We have made changes based on the suggestions and believe the writing to be much improved. In particular, we appreciate the feedback on the description of other firn models compared to SNOWPACK. We thank the referee for their input and hope that our modifications have addressed their concerns. Please find below our responses to comments in blue text, and specific changes made to the text in ***blue, bold, italic font.***
Line numbers refer to the original manuscript.

Major comments l. 71: "SNOWPACK uses near-surface atmospheric conditions as input". This is confusing; I assume more
input parameters are required to calculate the surface mass and energy balance at the upper boundary of the firn model? Such as surface radiation and mass fluxes? Upon further reading this is specified in l. 92, but please correct here to avoid confusion.

We have changed this to ***SNOWPACK uses a meteorological dataset as input*** to improve the accuracy
of this statement.

l.77: "Unlike semi-empirical firn models used in similar studies (e.g., Kuipers Munneke et al., 2015), SNOWPACK does not use a positive degree day parameterization for calculating melt." As it is written here, this is incorrect. The "semi-empirical" refers to the way snow densification is parameterised and does not refer to the way these models are forced. The study of Kuipers Munneke et al (2015) cited here
used an idealized/simplified melt forcing simply to facilitate interpretation of the development of firn aquifers. Other studies by that same author use realistic surface mass and energy balance to drive the firn model, i.e. equivalent to what has been done here. There remains an important difference though. In the model setting used here, SNOWPACK calculates its own surface energy balance, whereas in most
other studies the firn models are directly forced by surface skin temperature from regional climate models, being the result of the closure of the surface energy balance. Although not critical for this study, it is good to clearly separate these different approaches.

We appreciate this feedback and acknowledge that our text as written is misleading. In our effort to
distinguish SNOWPACK from other models, we have incorrectly made assumptions about other work. We have changed this sentence to the following:

***While many other firn models rely on surface skin temperature from the atmospheric forcing to calculate melt (e.g., Steger et al., 2017; Medley et al., 2022), SNOWPACK does not take this***
***approach.***

Section 2.3. The model coverage of Greenland is at relatively low resolution, and points that do not meet the spin up criteria are removed. As a result, only 1724 locations remain, an order of magnitude less than when the ice sheet would have been resolved at e.g. 10 x 10 km grid cells. This low resolution is also
apparent from e.g. Fig. 1. Could you comment at how that potentially influences the (ice sheet integrated) results, and also regionally especially in regions with strong climate gradients? And can you indicate in Fig. 1 the outline of the contemporary ice sheet, so that it becomes clear which parts of the ablation zone have been removed from the analysis?

This is a great point and something that we have considered but had not added to the text. Most of the grid cells excluded with the spin up conditions are in the lowest elevations of the ablation zone. Based on our results, we expect very little change in firn air content to occur in those areas simply because there is no firn there, but rather exposed bare ice. The original Figure 7f is a great example of how very minimal changes occur in firn air content where the initial temperature and melt rates are already very high. As such, excluding the very margins is likely to have very little impact on our results since the margins contain little to no firn. Even 1 °C of cooling is not likely to have a substantial impact there (Fig. 7f). We have made the following changes in Section 3.1 to help convey the point that the very edges have no firn:

*The largest changes in both warming and cooling experiments occur along the margins of the ice sheet but slightly inland from the edge of the model domain since firn is not present at the lowest elevations.*

We appreciate the suggestion for adding the ice sheet outline and have done so for the original Figures 1, 2, 3, and 9. In the captions, we have also added the following text to describe the outline: *The thin black line represents the ice sheet outline.* As an example of this figure modification, below is the modified Figure 1 and its caption.

[Figure]

*Figure 1: Greenland Ice Sheet mean annual (a) temperature, (b) snowfall, (c) melt, and (d) firn air content (FAC) from the 100-yr control climate (i.e., the 1991 climate) prior to the warming and cooling experiments. The thin black line represents the ice sheet outline.*

As for the spatial resolution, this is something we cannot change if we choose to continue using MERRA-2. Our results are limited by the native resolution of our chosen forcing. We selected MERRA-2 because it has been used before in a study with a more realistic model framework (Thompson-Munson et al., 2023). A finer resolution could offer more detail in the steeper sloped marginal areas (e.g., the southeast), and it would be interesting (though beyond the scope of this work) to use an alternative atmospheric forcing such as RACMO.

Same section and results section: you select a single year as a baseline for all your experiments. Am I correct that Figs. 1a, b, c then represent the climate of 1991? Apart from the absence of a trend in firn air content for that year (is this averaged over the ice sheet, how does it hold regionally?), can you also provide information on the temperature/accumulation/melt of that particular year relative to climatology? It appears from Fig. 1 that melt is rather low along the western marginal ice sheet (although I realise that most of the ablation zone is not part of the model domain).

Yes, that is correct that this represents the 1991 climate. We have added *(i.e., the 1991 climate)* to the Figure 1 caption and added *(i.e., for the 1991 climate)* to line 121 to clarify this point. The addition of the ice sheet outline should help demonstrate that yes, the melt on the western margin is low because the ablation zone is excluded with the spin-up procedure. However, we have also updated the figure in response to another review, and it now shows some of the melt along the western margin—though not within the ablation zone. Please see the modified Figure 1 in the above response.

As for 1991 relative to the climatology, we would point you to Thompson-Munson et al. (2023) Figure A1 which shows time series of MERRA-2 variables used to force SNOWPACK from 1980 through 2020.

Forcing experiments: if I understand correctly, only 2 m air temperature is varied, but other variables that normally are closely connected to air temperature (incoming longwave radiation, specific humidity, rainfall fraction) are left unchanged? If so, please mention this specifically to avoid confusion.

We have specified "**2-m**" in line 117 and in added the following at line 118:

*Note that we do not alter any other forcing variables and we use "air temperature" to mean 2-m air temperature throughout the rest of the manuscript.*

Fig. 7: What causes the fluctuations with frequency of about 1/decade in panels a and c?

These fluctuations are a result of the output frequency of the model combined with leap days. They are not "real" signals within the data but rather just a result of the 14-day model output frequency being affected by an extra calendar day every 4 years. We double checked this in a few examples by changing
the output frequency to hourly and then resampling to a 14-day frequency. We chose to output data on a coarser temporal resolution than the model is run at in order to keep file sizes manageable. Also, to explain this point in the manuscript, we added the following to the original Figure 7's caption:

*Also note that the approximately decadal oscillations in the time series are a result of leap years*
*affecting the 14-day model output frequency and are not real signals within the data.*

Minor and/or textual comments l. 20: asymmetric -> temporally asymmetric (to distinguish it from spatially asymmetric, which I thought
was meant when first reading this sentence)

We have changed this here and in the abstract in line 17.

l. 25: drives meltwater runoff ->  drives enhanced meltwater runoff
Done.

l. 31: limit -> limit and/or delay (in the case of aquifers)

Done.

l. 32 and following discussion: Firn air content and density are presented here as two separate characteristics, but it would seem to me that they are one-to-one coupled? Or am I overlooking something?
This is a great point that another reviewer also brought up (see response to Referee #2, line 116). We have made the following changes to this paragraph:

"such as air content and density" → *like air content*
"Changes in firn air content and firn density" → *Changes in firn air content*

We kept one distinction between density and firn air content because we refer to specific papers that looked at one or the other variable.

l. 139: "slightly inland from the ice edge". As large parts of ablation zone are not considered, the ice edge can be >100 km away. Perhaps better use "slightly inland from the equilibrium line".

We have changed this to:

*slightly inland from the edge of the model domain since firn is not present at the lowest elevations.*

---

## Author Comment (AC3)

**REFEREE 3: ERIN PETTIT**

This work presents and overview of how Greenland's firn porosity might change under slight warming or cooling scenarios.

The modeling and results are insightful and interesting – I really just want more! But in a shorter, more concise structure, and figures that really draw the reader to the main points. I agree with several of the points of the other reviewer – showing the model reaching equilibrium would be great. And testing another magnitude of change. But I realize that is effort. Also, as a physicist, I really want to see the results tied back to the physics of firn, since this is what would allow us to take these results and transfer them to other places…

We thank the referee for their thoughtful feedback and insightful comments that have helped us improve the manuscript. In particular, we appreciate the suggestions to make the writing more focused and specific. We have addressed the comments below and have noted the changes we have made to the manuscript text and figures. Please find below our responses to comments in blue text, and specific changes made to the text in *blue, bold, italic font.* Line numbers refer to the original manuscript.

General comments:

- the abstract suggests that the work will focus on the physics of the nonlinear relationships, then discuss this in the context of Greenland, but the paper jumps immediately into the large scale integrated Greenland response without showing much of the physical reasoning. For example, how much does a steady trend in air temperature affect the firn compared to an individual warm event (one hot summer, for example)?

Regarding the abstract, we noted our modifications in specific comments below (see lines 88, 134 of this document). We have also made the abstract more specified and precise by modifying lines 9-10 to say:

*To better understand the nature and timescales of firn's response to air temperature change on the Greenland Ice Sheet, we use a physics-based model to assess the effects of atmospheric warming and cooling on firn air content in idealized experiments.*

Regarding the questions and suggestions for new experiments (e.g., trend, extreme events) posed by the reviewer, we agree that they are very interesting and likely could be answered with a tool like SNOWPACK. However, they are beyond the scope of this work. For this work - we take a simple and reasonable approach (1 °C temperature perturbations) that allows us to place our results in the context of previous research (Kuipers Munneke et al., 2015) but add new exploration of idealized temperature–firn relationships over the entire Greenland Ice Sheet. Additionally, we aim to isolate the system's response to just one variable. Our experimental design adds new insights on timescales and magnitudes of response that make physical sense. We agree more work in this idealized framework could provide additional insights, and fit well within the research efforts of the firn-climate community.

- at the end of the abstract/introduction, I was still trying to figure out where this paper was headed. It was unclear to me what the goals of the paper were and what methods would be used or what the findings were (other than big general statements). It would be nice to be able to extract from the abstract and introduction what is really novel about what this paper is offering and how it is getting there. I think this could be helped by just making each sentence, each paragraph a bit more specific (not longer, just more specific).

We appreciate this suggestion and we believe the changes we have made in response to other comments have resolved this issue (see lines 27-32 and 134 of this document)

- The methods section is a bit limiting from my perspective. Perhaps this is because I like to know what physics is going on! More information is provided about the surface boundary condition than to the internal physics of the firn model - a summary of the key assumptions made by the model would be really

helpful. The model also uses the same parameters as a previous paper (but I don't have time to go read that paper) and then is compared to the results from that paper to suggest that the model is working well. I'm a bit confused as it seems to me that the statement implying validation (Line 133) of the model is just comparing the model to itself.

In this work, we are building on previous work using this exact model set-up. Thus, an in-depth model description including assumptions and physics included is not needed as this exact model that has been used and described (Bartelt and Lehning, 2002; Lehning et al., 2002a, b), including for the Greenland Ice Sheet (e.g., Thompson-Munson et al., 2023). As a result, we have aimed to provide enough model description for readers in Section 2.1. In this section, we provide a list of studies that have used SNOWPACK (lines 67-70), information about how the model calculates density for new layers within a Lagrangian framework, and how melt is calculated with a surface energy balance scheme. We also provide key parameters here that could be applied to other firn models (e.g., the bucket scheme, the spin-up period) in Section 2.3. Please see lines 161-167 of this document for further comments.

In line 133, we agree with the reviewer that the original text was confusing. Our intention was to show that our modeling framework and choice of 1991 made sense by falling within the range of previous simulations using a more realistic climate. As such, we have changed this sentence to the following:

***We ensure that the idealized atmospheric forcing can produce firn air content values within the range of those previously simulated, and we find that the mean of 19.9±8.3 m and spatially integrated firn air content of 35621 km$^3$ are consistent with results from the SNOWPACK study that uses the same parameters over Greenland (Thompson-Munson et al., 2023).***

- Also related to methods - the goal of the paper implied trying to figure out how firn responds. But given that the results are only provided in the context of large scale changes on Greenland, I don't know how to take the results and apply them to a mountain glacier or to Antarctica. I would suggest an approach of sweeping through a range of parameters and looking at 1D results for each, to show the changes in the snowpack that are possible. Or at a minimum, spending much more time on results such as Fig 7/8 - where a few modeled points with different characteristics are compared.

This paper and the results in it are Greenland-focused; we specifically want to know how *Greenland* firn responds. As a result, taking these results and applying them to a mountain glacier or Antarctica may not be appropriate and also is beyond the scope of this work. We agree it is helpful to clarify this goal. We have added the word "***Greenland***" to several sentences throughout the paper (namely in the abstract, discussion, and conclusion) to clarify our intentions. Additionally, there are certainly implications of this work that could be applied to other ice sheets or glaciers, but fully diving into those and describing them is outside the scope of the present study. We acknowledge that your suggestions could make a very interesting additional paper, but we hope that this manuscript lays the groundwork for further research into more general atmosphere–firn interactions.

- this paper is long and quite wordy in ways that I don't think are necessary to get the message across. Many general statements that could be deleted or condensed. Figures made more focused on the

We have made multiple revisions to improve clarity and reduce length.

- There are a lot of figures that to me seems to re-inforce the same conclusions and I got a bit overwhelmed trying to figure out what new I was learning from each figure. Each figure should have a clear message that is easy to grasp even if the reader is skimming the papers.

Thank you for expressing this concern. We selected figures to fully show our results. We use the text to describe what is in each figure and why it is important. That said, we understand this concern. In response, we reduced the number of figures in the main text. Specifically, we have moved the original Figure 6 to the appendix. This figure reduction enables us to communicate our results with fewer figures. Figure numbers have been updated to reflect this change and the main text now includes only 9 figures while the appendix has 2.

115 - I really like the idea that this asymmetry might affect the way the ice sheet grows or shrinks, but I feel like that message got a little lost along the way.

We're very glad to know this idea was appreciated. We believe this is a powerful implication of our results, but we do not believe the data support making overly strong claims about the role of firn in ice sheet growth/decay. An interesting next study could be to attempt to quantify the contribution of firn air content 120 changes to overall ice sheet mass changes.

Specific comments:

Abstract:

125
- In the first sentence I'd suggest saying that the porous layer of snow stores meltwater and limits **the rate at which Greenland contributes to sea level rise**.

Done.
130
- the abstract states that the the paper demonstrates that fir air content is more efficiently depleted… but doesn't provide any information to the reader as to the specific findings.

We have added the following sentence at line 12 to provide more specific findings while keeping the focus 135 on the big-picture results.

*One hundred years after a 1 °C temperature perturbation, warming decreases air content by 9.7 % and cooling increases it by 8.3 %.*

140 Introduction

Line 24 - check wording, should be contribute, instead of contribution

Fixed.
145
Lines 32-40 - I understand the authors are trying to be succinct, but much of this paragraph is written in fairly general statements and the citations do not seem appropriate. For example the idea that meltwater changes firn was discovered much before 2015. If the authors wish to cite the more recent papers, then please be specific about what new knowledge those newer papers contributed.
150
Following this suggestion and those of another reviewer, we have removed "(MacFerrin et al., 2022)" in reference to general statements about overburden stress and "(Kuipers Munneke et al., 2015)" in reference to meltwater altering firn properties.

155 Methods

The model - In general I would like to seem more details. The reader should be able to read this paper and get at least the basic elements of the model the parameters used and any necessary details without having to go read another paper or papers first.
160
Given the length of the paper and the goal of remaining on message, we chose to only describe the general workings of the model. Several other papers have used and thoroughly described the SNOWPACK model on ice sheets and specifically on Greenland, and we have cited those papers in lines 67-70. In the present study, we use SNOWPACK as a tool to help understand firn–climate interactions. 165 Other firn models should be able to reproduce similar results, so detailing the inner workings of the chosen model would not contribute to the present focus of the paper. See lines 62-70 of this document for more.

The experiments - Line 100 - "we require" - why? I see why you want no trends, but our real world does have noise at all timescale, so even in a steady climate there will be noise. I would suggest repeating the experiments with constant climate but added noise. Perhaps that is a different paper (if the authors want to contact me for more information on this please do, I've done some not-yet- published work on firn thermal structure in noisy versus "pure" steady climate conditions).

This study builds off existing work such as that of Kuipers Munneke et al. (2015). While we do not use the same model or even the same forcing, we appreciated their study design that used an idealized forcing. As such, we chose to take a similar approach in the creation of our forcing data. We acknowledge your point that the real world does have noise at all time scales, and that is certainly an idea that we discussed in the design of this study. However, we came to the same conclusion that you did: it could serve as a study on its own. For this paper, we chose to start as simply and reasonably as possible and explain firn responses in an idealized framework. We also note that since we used a full year of MERRA-2 data, there is noise included on the sub-annual scale. We also found our overall results are not sensitive to the year of MERRA-2 data we use to define the base climate.

Line 104-105 - We verified - how? What do you consider negligible? I am assuming you ran the same model using a different year's climate and the results on warming/cooling were less than 1% different or something?

We ran the model for additional years and found that warming always has a slightly stronger (a few percent) effect than cooling. The exact firn air content values changed, but the asymmetric relationship remained. 1991 is a good choice of year because there were no significant changes to the firn air content that would affect our spin-up period. As an example, we used data from 1980 (the first year available) and created the following figure in our preliminary work. We found that the same warming/cooling relationship existed, but firn air content increased during the year of 1980, which resulted in a trend in the control data. We avoided this trend by using a year with little change (1991).

[Figure]

Finally some of the limitations of the model and the experimental design are not noted until the very end of the discussion section - and it seems like it would be nice to put those right up in the methods because they are choices that have been made in designing the experiment and the analysis.

We have chosen to keep the study limitations in the discussion so that we can discuss them in the context of our results and other papers. Moving this text to the methods would introduce topics that do not appear until later in the study, and we would prefer to not split up the discussion of limitations.

Results

210 Lines 120-135 - this entire paragraph is presented as results, but to me it seems like a confirmation that the model mostly does what we expect. Maybe just rewording this paragraph? statements like "melt occurs where the temperatures are highest" seems so obvious that I wonder what the authors want me to learn from this paragraph.

215 Thank you for pointing this out. As you noted, we did intend for this paragraph to show that the model behaves as expected. However, we also included it to make the results approachable to a wider audience that may be less familiar with ice–atmosphere interactions. To give this paragraph more context, we have changed the first sentence to:

220 **We first present the modeled ice sheet and climate mean state during the control period prior to any atmospheric warming or cooling to confirm the model behaves as expected (Fig. 1).**

We also removed the following sentence as we agree that it adds very little new information:

**During the control period, melt occurs where the temperatures are highest.**

225

Line 136-139 - these statements could use specifics. "Idealized warming causes depletion of firn air content" - by how much, how measured. The first specifics offered are for the total spatially integrated air content.

230 We have changed this sentence to reflect that we mean everywhere on the ice sheet. It now reads:

**When perturbed from the control state (Fig. 1), idealized warming causes depletion of firn air content (Fig. 2a) while idealized cooling causes generation of firn air content (Fig. 2b) in all modeled grid cells in Greenland.**

235

I think overall this section could be condensed down and focused on comparing what was learned to what was expected.

We have condensed this section down by removing the last two sentences about reaching equilibrium.

240

Line 151 - timescales of response to a perturbation are typically defined by an e-folding time, I'm not quite sure I understand the need to use the percentages.

We chose to use percentages because using an e-folding time assumes the data behave exponentially.
245 While the ice-sheet-integrated curves appear to be related exponentially/logarithmically, many individual grid cell time series are not. The original Figure 7 provides a few examples (namely, panels e and f) of where assuming an exponential change in firn air content would be a poor choice. The times to percent changes instead do not make any assumptions about the shape of the curves and are more robust quantifications of the rates of change.

250

Moreover, the equation that would best fit the data in, say, Figure 2 (the ice-sheet-integrated firn air content) would be in the form $y = A - Ae^{-t/\tau}$ where $\tau$ is the e-folding time. The data do not reach an equilibrium within the experiment time frame, so we would have to make assumptions about the value of A to get at an e-folding time.

255

Line 167-169 - again very general statements.

We have removed these statements.

260 Figures

- most of the figures really could use contours instead of just colors. I can't tell dark from darker colors and the authors point to various areas where they expect us to see color variations (such as SE

Greenland coastal areas). Also - many of the figures contain the full spinup period - this seems like a lot of wasted space, perhaps modify it to zoom in on the results that are the most useful for telling the story.

We appreciate this suggestion and have made modifications to several figures to address this. We have updated Figures 1, 2, and 3 using filled contours rather than smooth shading. (Since the original Figure 9 shows discrete categories of responses that are spatially unrelated, we have not added contours here.) As an example of our updates, below is the modified Figure 2 and its caption.

[Figure]

**Figure 2: Change in firn air content (FAC) for the Greenland Ice Sheet calculated as the final firn air content minus the mean of the control conditions for (a) warming by 1 °C and (b) cooling by 1 °C. The thin black line represents the ice sheet outline. (c) Time series of the firn air content volume anomaly integrated over the full ice sheet for the control (gray line), warming (red line), and cooling (blue line) periods. The control period and cooling experiment use the positive values on the left y-axis and the warming experiment uses the negative values on the right values on the right y-axis. (d) Changes in spatially integrated firn air content in km³ at the end of the experiments partitioned into wet firn (melt is present) and dry firn (melt is not present) areas for warming (red) and cooling (blue).**

While we acknowledge that showing the spin-up period in the figures takes up room, we are choosing to show it in order to support our claim that the control period is stable. It can be valuable for equal-length time periods to be shown when comparing the perturbed states to the control state, as it makes for a more direct comparison.

- is there a reason not to use an e-folding time to express the timescales, Figure 3, for example, could be just one plot instead of 6. Or maybe 2 if the authors want to emphasize how much change happens in the first 2.5% of time.

Please see our earlier response regarding e-folding times, as we believe we have addressed this comment in that response (lines 244-254 of this document).

- Among figures 7 an 8, a few things could make the data more digestible - In fig 7 for example, All 6 could be plotted on the same curve as relative change in FAC. Because of the different scales, it is hard to compare them. Also Fig8 take a while to digest - similarly the "control climate" takes up a lot of ink, and then what my mind really wants to see is a diagram what a firn core might look like in each situation. Also Fig 8 only shows two locations - it would be helpful to find a way to express this information for all of them.

The original Figures 7 and 8 do contain a substantial amount of information, but we feel that it is necessary for supporting our conclusion that the temperature–firn relationship is highly complex and to

show all of the physics that is included in this modeling (as summarized in the original Figure 10). Plotting all six examples from the original Figure 7 on the same curve (even if normalized) would result a loss of valuable information and arguably a more confusing figure. For instance, plotting panels (e) and (f) on the same axis would result in (f) being very hard to see since ~3 m of change occurs in (f) while ~8 m occurs in (e). The intention of this figure is not to directly compare each of the examples, but rather compare the signature of warming and cooling *within* each panel and showcase the wide variety of responses we find in just 6 out of 1724 cases.

As for Figure 8, we have in part addressed this in a previous comment, but we also acknowledge that a similar type of figure is shown in Kuipers Munneke et al. (2015) and we found theirs to be very informative. Finally, we use this figure to show one of the simpler cases and one of the most complex cases as end member responses. While the other example could be interesting, there are 1724 of these panels we could also plot and could be potentially interesting. We are choosing to just show two examples that help drive home our conclusions instead. However, we have supplied all the code and data (see the code/data availability section) so readers are able to explore the full dataset.

Again - lots to like in this paper, most of my comments are towards focusing the message and making it easier to read and digest.

Erin Pettit

**References used in this response**

Bartelt, P. and Lehning, M.: A physical SNOWPACK model for the Swiss avalanche warning Part I: numerical model, Cold Reg. Sci. Technol., 23, 2002.

Kuipers Munneke, P., Ligtenberg, S. R. M., Suder, E. A., and Van den Broeke, M. R.: A model study of the response of dry and wet firn to climate change, Ann. Glaciol., 56, 1–8, https://doi.org/10.3189/2015AoG70A994, 2015.

Lehning, M., Bartelt, P., Brown, B., Fierz, C., and Satyawali, P.: A physical SNOWPACK model for the Swiss avalanche warning Part II. Snow microstructure, Cold Reg. Sci. Technol., 21, 2002a.

Lehning, M., Bartelt, P., Brown, B., and Fierz, C.: A physical SNOWPACK model for the Swiss avalanche warning Part III: meteorological forcing, thin layer formation and evaluation, Cold Reg. Sci. Technol., 16, 2002b.

Thompson-Munson, M., Wever, N., Stevens, C. M., Lenaerts, J. T. M., and Medley, B.: An evaluation of a physics-based firn model and a semi-empirical firn model across the Greenland Ice Sheet (1980–2020), The Cryosphere, 17, 2185–2209, https://doi.org/10.5194/tc-17-2185-2023, 2023.

---

## Author Response (AR2)

REVIEWER #2

We are very thankful to this referee for their insightful feedback that has improved the accuracy and clarity of this manuscript. We especially appreciate the time taken to review it a second time and clearly explain their thoughts. Our responses to the below comments can be found in blue text.

Thanks for the answer and revision, but the revised text is still incorrect: one cannot calculate melt rate from surface temperature, as the latter is constant during melt. To calculate melt rate, the full surface energy balance must be known or some approximation (degree days) must be used.

Independent of the firn model physics, to force any firn model at its upper boundary, one needs surface temperature, surface accumulation and surface liquid water flux (melt and rain). These can be obtained as follows:

1) Prescribe all from a model or observations.

2) Prescribe (observed, modelled) surface accumulation, rain, near-surface meteorology and radiation fluxes (T2m, V10m, SWnet, LWin), and then use those to close the surface energy balance, which yields surface temperature and melt rate (if the surface is at melting). If no radiation data are handy, a simplified (degree day) method can be used to estimate melt.

If I remember correctly, in SNOWPACK there is a switch to choose between these options.

We appreciate this comment and thank the referee for clearly explaining this. We apologize for not correctly understanding the original comment. In an effort to make this section accurate and clear, we have simply removed the problematic phrase. It was originally incorporated to set SNOWPACK apart from other models. However, we have removed it and are now keeping this section solely focused on SNOWPACK.

The following has been removed "While many other firn models rely on surface skin temperature from the atmospheric forcing to calculate melt (e.g., Steger et al., 2017; Medley et al., 2022), SNOWPACK does not take this approach. Instead".

Review of Thompson-Munson paper on Greenland's Firn

40  I really appreciate all the effort the authors went to to revise the paper acknowledging the reviewers comments. Specially, I appreciate that I better understand the overarching goal of the paper - and that is now more clearly communicated in the new draft.

The paper reads much more smoothly now and I understand why some decisions were made.
45

I have just a few additional specific comments:

We are very thankful to this referee for their insightful feedback that has improved the manuscript. We especially appreciate the time taken to review it a second time. Several
50  changes have been made, and our responses to comments can be found below in blue text.

Abstract
Line 11 - There are a few more places where it can be made even more clear that the intent is specifically to study Greenlands firn air content. "warming and cooling on *Greenland's*firn air
55  content in an idealized *climate* experiment" - when I think of the phrase idealized model, I think of idealized geometry, idealized parameter spaces, etc. So it is helpful to the reader to be clear that in what way this is idealized.

Thank you for this suggestion. We have made the recommended changes in line 11.
60

Line 13 - "warming decreases the *integrated* air content… "

We have added "spatially integrated" to this phrase.

65  Line 15 - dependence (not y)

Done.

Intro
70  Line 57 - *Greenland's* firn behavior

We have changed this phrase to "Greenland firn's behavior"

Line 63/64 - either use pore-space loss (hyphenated). Or "loss in pore space" I prefer the latter -
75  also "gain in pore space"

To be concise, we have added hyphens per this suggestion. This sentence now reads as: "Specifically, temperature–firn interactions amplify pore-space loss more in a warming climate than they amplify pore-space gain in a cooling climate"
80

2.1 Model Description
The model description still can use a few more elements. Some variation stating that it is a 1-d conservation of energy and mass, Lagrangian framework. The thermal conductivity for each layer is based on ?? varies with density/crystal structure?? (which isn't stated here, but is
85  referred to later in line 273)… Specifically, I don't know what this statement means: "uses an energy balance model to calculate melt in a way that incorporates processes occurring

throughout the firn column in addition to those at the surface" does that just mean that energy is conserved within/across each layer in the model (i.e. a 1D conservation of energy model)?

Thank you for this feedback. We have made changes to section 2.1 in response to these comments. In particular, we have elaborated on the description and clarified the language where appropriate. SNOWPACK is a complex model that is introduced across three papers (Bartelt and Lehning, 2002; Lehning et al., 2002a, b), and is challenging to fully describe within this manuscript. We have followed what other authors have done in recent SNOWPACK papers (e.g., Dunmire et al., 2024; Banwell et al., 2023) and provided an overview of the model without getting into the fine details that detract from the model results. With this approach and the great suggestions from this referee, we hope that section 2.1 is easier to understand and more informative to the reader. Below, please find the specific changes we have made to this section.

In line 72, we changed "SNOWPACK" to "This one-dimensional model" in order to restate that it is a "single-column" model (line 68).

In line 75, we added the following to address the mass/energy conservation comment: ", and SNOWPACK solves the partial differential equations that describe mass, energy, and momentum conservation".

We added the following sentence at the end of this paragraph (line 77) to address the thermal conductivity: "Snow and firn microstructure governs physical properties like the thermal conductivity and is captured in the model's description of grain radius, bond radius, sphericity, and dendricity (Lehning et al., 2002b)."

We have changed the confusing sentence to explicitly mention the processes we alluded to. Line 80 now reads: "It uses this energy balance model to calculate melt in a way that incorporates several processes, including accumulation, snow-albedo feedback, percolation, and latent heat release (Wever et al., 2014, 2015, 2016)."

References:

Dunmire, D., Wever, N., Banwell, A.F., Lenaerts, J.T.M. (2024) Antarctic-wide ice-shelf firn emulation reveals robust future firn air content depletion signal for the Antarctic Peninsula. *Commun Earth Environ* **5**, 100 (2024). https://doi.org/10.1038/s43247-024-01255-4

Banwell, A. F., Wever, N., Dunmire, D., & Picard, G. (2023). Quantifying Antarctic-wide ice-shelf surface melt volume using microwave and firn model data: 1980 to 2021. Geophysical Research Letters, 50, e2023GL102744. https://doi. org/10.1029/2023GL102744

Finally, the one assumption that is left out until the discussion that I think is important for the reader to know up front is the assumption that all pore space is available for water, that ice lenses are not altering the pore space availability. No need to discuss it more in this section, keep the discussion of limitations at the end. But as it is an assumption of model, I think it needs to be here.

This is a great point that we agree is important to include early on. We have added it to the end of the methods in line 124: "We assess how air temperature perturbations impact the firn air content, which we calculate in the same manner as in Thompson-Munson et al. (2023) and assume that all pore space is available for storing meltwater."

The domain, boundary conditions, and initial spin up is explained well.

140 Sec 3.3
Line 214 - "the mean summer air temperatures in all three experiments are below 0" - I see that the final experiment (f) is above zero, not below.

Thank you for noting this. By "all three experiments", we were referring to the control, warming,
145 and cooling experiments in panel (d) alone. We now see that this is very confusing so we have changed it to "the mean summer air temperature throughout the 200 years is below 0°C".

Discussion
To make this clearer I'd suggest enumerating the major pathways through which air temp can
150 alter firn air content:
1. Compaction rate of dry firn
2. Increasing the bulk thermal conductivity
3. Melting fills pore space

155 We find this to be a great method for introducing these important processes. We have added the following at line 269: "We have identified three categories of processes altering firn air content: (1) dry firn compaction (Fig. 9a, b), (2) thermal property changes (Fig. 9c, g, h), and (3) meltwater production (Fig. 9d, e, f)."

160 Line 276-277 - The sentence starting "Increasing the air temperatures…" doesn't seem like it is necessary, the sentence afterwards seems to explain the process sufficiently. I'd suggest cutting the sentence.

We have decided to keep this sentence because it introduces the pivotal idea that the
165 relationship is nonlinear. This is an important characteristic of the relationship because it partially explains the asymmetric response to warming and cooling.

Line 295 - "latent heat *from* freezing" (the "latent heat of freezing" is a constant 334kJ/kg)

170 Thank you for this correction. We have changed this to "latent heat released from refreezing." We have also made this change where it appears again in line 305.

Lines 320-340 seems like they belong in the conclusion section.

175 We appreciate this suggestion. However, since these paragraphs contain references to other work, we feel they belong better in the discussion.